# A comprehensive numerical procedure for high-intensity focused ultrasound ablation of breast tumour on an anatomically realistic breast phantom

**Reza Rahpeima**[ID], **Chao-An Lin**[ID]*

Department of Power Mechanical Engineering, National Tsing Hua University, Hsinchu, Taiwan

* calin@pme.nthu.edu.tw

**Data Availability Statement:** All relevant data are within the manuscript and its Supporting Information files

## Abstract

High-Intensity Focused Ultrasound (HIFU) as a promising and impactful modality for breast tumor ablation, entails the precise focalization of high-intensity ultrasonic waves onto the tumor site, culminating in the generation of extreme heat, thus ablation of malignant tissues. In this paper, a comprehensive three-dimensional (3D) Finite Element Method (FEM)-based numerical procedure is introduced, which provides exceptional capacity for simulating the intricate multiphysics phenomena associated with HIFU. Furthermore, the application of numerical procedures to an anatomically realistic breast phantom (ARBP) has not been explored before. The integrity of the present numerical procedure has been established through rigorous validation, incorporating comparative assessments with previous two-dimensional (2D) simulations and empirical data. For ARBP ablation, the administration of a 0.1 MPa pressure input pulse at a frequency of 1.5 MHz, sustained at the focal point for 10 seconds, manifests an ensuing temperature elevation to 80°C. It is noteworthy that, in contrast, the prior 2D simulation using a 2D phantom geometry reached just 72°C temperature under the identical treatment regimen, underscoring the insufficiency of 2D models, ascribed to their inherent limitations in spatially representing acoustic energy, which compromises their overall effectiveness. To underscore the versatility of this numerical platform, a simulation of a more clinically relevant HIFU therapy procedure has been conducted. This scenario involves the repositioning of the ultrasound focal point to three separate lesions, each spaced at 3 mm intervals, with ultrasound exposure durations of 6 seconds each and a 5-second interval for movement between focal points. This approach resulted in a more uniform high-temperature distribution at different areas of the tumour, leading to the ablation of almost all parts of the tumour, including its verges. In the end, the effects of different abnormal tissue shapes are investigated briefly as well. For solid mass tumors, 67.67% was successfully ablated with one lesion, while rim-enhancing tumors showed only 34.48% ablation and non-mass enhancement tumors exhibited 20.32% ablation, underscoring the need for multiple lesions and tailored treatment plans for more complex cases.

**Funding:** The author(s) received no specific funding for this work.

**Competing interests:** The authors have declared that no competing interests exist.

**Abbreviations:** qd, dipole domain source, N m-3; R, universal gas constant, J K−1 mol−1; A, frequency factor, s-1; Re (), real part of a complex number; c, speed of sound, m s-1; T, temperature, K; $c_p$, specific heat in constant pressure, J kg-1 K-1; $T_b$, arterial blood temperature, K; $C_{p,b}$, specific heat of blood, J kg-1 K-1; $T_\infty$, ambient temperature, K; f, cyclic frequency, s-1; t, time, s; h, heat convection coefficient, W m-2 K-1; v, acoustic particle velocity vector, m s-1; I, acoustic intensity magnitude, W m-2; i, imaginary unit Greek symbols; k, wave number, m-1; α, degree of tissue injury; kh, heat conduction coefficient, W m-1 K-1; $α_a$, attenuation coefficients, dB m-1 MHz-1; n, polynomial order of the Arrhenius equation; ΔE, activation energy, J mol-1; n→, surface normal vector; ε, surface emissivity; p, total pressure, Pa; $θ_d$, fraction of necrotic tissue; $p_0$, input pressure signal amplitude, Pa; λ, acoustic wavelength, m; $p_A$, amplitude of pressure oscillation at the pressure anti-node, Pa; ρ, density, kg m-3; $Q_m$, monopole domain source, s-2; $ρ_b$, density of blood, kg m-3; $Q_{met}$, metabolic heat generation rate, J s-1; $σ_{SB}$, Stefan-Boltzmann constant, kg s-3 K-4; $Q_s$, acoustic heat source rate, J s-1; ω, angular frequency, rad s-1; q→, heat transfer vector, J s-1; $ω_b$, blood perfusion rate, s-1.

# 1 Introduction

According to the World Health Organization (WHO), the incidence of cancer has doubled over the past three decades and is expected to triple by 2030 [1]. Breast cancer is one of the most prevalent forms of cancer today, posing a significant threat to the lives of individuals of all genders each year. In 2021, the American Cancer Society reported 284,200 new cases of breast cancer and 44,130 deaths in the United States alone [2]. Although surgical treatment for breast cancer has been successful, it is highly invasive, particularly for early-stage malignancies such as small tumours. As a result, there has been a growing need for less invasive therapies for breast cancer in recent years [3].

High-intensity focused ultrasound (HIFU) is a procedure that utilizes high-frequency sound waves to eradicate cancer cells. This approach employs a focused ultrasound beam that is directed toward a specific area affected by cancer. Upon contact with the cell, the high-intensity ultrasonic beam triggers a rapid increase in temperature, which leads to the death of the cell. HIFU has demonstrated promising results in clinical trials for the treatment of both benign and malignant tissues [3, 4]. Although there are alternative thermal ablation techniques such as laser [5], microwave [6], and magnetic fluid hyperthermia (MFH) [7].

One of the well-known alternatives is Magnetic Fluid Hyperthermia (MFH), for which we previously proposed a comprehensive simulation platform for the ablation of breast tumors [8]. Both HIFU and MFH have unique advantages and disadvantages in the context of breast cancer ablation. The choice between them depends on the specific characteristics of the tumor, patient condition, and the available resources and expertise. But briefly, HIFU offers more precise targeting and real-time monitoring during the procedure, but it is limited by penetration depth and has potential for skin burns. On the other hand, MFH provides better penetration for deep-seated tumors and compatibility with other therapies, but it faces challenges with nanoparticle delivery and control over heating effects.

HIFU is gaining popularity due to several factors. One of the main reasons is the widespread use of ultrasound in medicine. HIFU offers the advantage of precise focusing of acoustic waves on the body's deepest organs. It has also been found to be highly effective in disrupting and increasing the permeability of cell membranes. Moreover, HIFU is minimally invasive, nonviral in nature, low cost, and lacks ionization. Furthermore, it theoretically has the potential to treat a limitless number of cancers [9].

High-intensity focused ultrasound (HIFU) has been identified as a potential technique for the ablation of breast tumors. While Deckers et al. [10] highlighted the technical feasibility of a dedicated breast MR-HIFU system, their analysis was somewhat limited by the lack of a diverse patient cohort and did not fully explore the variability in treatment outcomes based on different breast densities or tumor locations. Peek et al. [11] provided a comprehensive review of HIFU applications, yet their discussion predominantly centered around technical descriptions without critically assessing the clinical efficacy and patient outcomes in a real-world setting. Feril et al. [12] while reviewing clinical applications, did not address the economic and infrastructural challenges of implementing HIFU widely in healthcare systems, which could have provided deeper insights into the practical barriers to its adoption.

Several academic studies have explored this technology, although most of them have been experimental in nature. A dearth of comprehensive numerical studies that can accurately model this complex multiphysics phenomenon on real phantoms still exists. In light of this, the present study aims to investigate this phenomenon numerically. To this end, we discuss some of the latest and most reputable analytical and numerical studies on this topic.

In 2018, Yoon and colleagues [13] proposed a multi-resolution simulation technique based on the finite-difference time-domain formulation to model the transcranial propagation of

acoustic waves from a single-element transducer. In the same year, Gupta et al. [14] conducted a numerical analysis of the thermal response of tissues under high-intensity focused ultrasound (HIFU) treatment. They utilized a 3D physical domain to represent a two-layered and multi-layered medium consisting of water and liver tissue and evaluated the pressure and temperature distribution in the medium. Their study significantly advances our understanding by demonstrating the complex interactions of HIFU with multi-layered biological tissues. However, their study was confined to liver tissue, and thus its applicability to breast tissue, which has different acoustic and thermal properties, remains untested. Rezaeian et al. [15] developed a numerical model to study HIFU-mediated intraperitoneal delivery of thermosensitive liposomal doxorubicin for cancer chemotherapy. In 2019, Montienthong and colleagues [3] conducted a computer simulation of HIFU ablation for breast cancer treatment. They adopted a simple 2D-axisymmetric model of the breast and simulated the pressure acoustics and heat transfer in the breast. Their approach while informative, oversimplifies the geometry of the breast, potentially leading to inaccurate predictions of HIFU therapy outcomes.

Furthermore, Gupta et al. [16] performed a non-Fourier transient thermal analysis of biological tissue phantoms subjected to HIFU. Their study focused on investigating the thermal wave model of the bioheat transfer equation (TWMBTE) instead of the Pennes bioheat transfer equation (PBHTE), which considers the finite propagation speed of the thermal front. In 2020, Sadeghi-Goughari et al. [17] presented an analytical and numerical model for HIFU enhanced with nanoparticles. Solovchuk et al. [18] carried out a computational study for investigating acoustic streaming and heating during focused ultrasound ablation of liver tumor. Further, Mohammadpour et al. [19] carried out a numerical analysis of heat transfer and hemodynamics of HIFU ablation in the porous liver. In 2021, Singh et al. conducted a numerical study to investigate the effectiveness of pulsed ultrasound in assisting thermo-therapy for the ablation of subsurface tumors, focusing on the temperature variations and thermal dose distribution achieved during the treatment [20]. Okita et al. [21] investigated the effects of breast structure on high-intensity focused ultrasound (HIFU) focal errors. They conducted numerical simulations of breast cancer HIFU ablation using digital breast phantoms constructed from MRI data of 12 patients. However, this study does not specifically show temperature distribution and the resulting amount of necrotized tissue. Instead, it emphasizes how different breast tissue structures can cause focal errors in HIFU treatments, highlighting the need for precise targeting and adjustment based on individual anatomical differences.

Our research addresses a significant gap in the existing literature concerning the simulation of HIFU for breast tumor ablation. While previous studies have primarily relied on 2D simulations, these models inadequately represent the three-dimensional complexities of breast anatomy and the interaction dynamics of acoustic waves within such a structure. Yadav et al. highlighted the transition from 2D to more advanced 3D models using k-wave simulation, underscoring the limitations of earlier 2D simulations in accurately representing complex breast anatomy and treatment dynamics [22]. Badawe et al. focused on the effect of pulsed HIFU on breast cancer and compared 2D and 3D simulation paradigms, pointing out the inadequacies of 2D models in capturing the three-dimensional complexities of breast tumor anatomy and treatment effects [23]. Recognizing this limitation, our study introduces an all-in-one 3D FEM-based simulation approach using an anatomically realistic breast phantom. This method not only promises greater accuracy in simulating the physical phenomena involved but also provides a more reliable predictive model for clinical outcomes. By integrating detailed anatomical data into our simulations, we aim to refine the precision of HIFU therapy, optimizing its efficacy and safety in clinical applications.

The development of an all-in-one simulation platform that models all physical aspects of HIFU treatment—including sound intensity, temperature achieved, and necrotized tissue—

using patient-specific breast models from medical images can serve as a powerful tool for pre-clinical evaluation and prediction. This platform addresses several key limitations and challenges in HIFU therapy. Traditional methods may struggle with accurately targeting tumors due to variability in patient anatomy [24], but patient-specific simulations enhance precision by accounting for individual anatomical differences, leading to more effective and safer treatments. The effectiveness of HIFU therapy depends on delivering the correct thermal dose to the tumor while sparing surrounding tissues, and realistic models predict thermal distribution more accurately, ensuring optimal dosing and reducing the risk of damage to healthy tissues [22]. Additionally, standard treatment protocols may not be suitable for all patients due to differences in tissue properties and tumor location [25]. Patient-specific simulations allow for tailored treatment planning, optimizing parameters such as intensity, duration, and focus of the ultrasound waves to maximize therapeutic outcomes. This comprehensive approach ultimately guides clinicians in choosing the optimal treatment dose and strategies.

Also, the 3D FEM simulation of acoustics, particularly in the context of HIFU, presents substantial computational demands [26]. To address this challenge, we have developed a simulation platform that enhances computational efficiency primarily through optimized mesh generation. This optimization is crucial for making the platform viable for clinical applications, enabling the effective prediction of outcomes for various HIFU treatment strategies and required dosages. Our approach aims to balance accuracy with computational expediency, thereby facilitating more practical and rapid clinical decision-making.

To obtain results as close to reality as possible, this study utilizes a physically realistic 3D model and an anatomically realistic breast phantom (ARBP). The ARBP is created from T1-weighted MRIs of patients in the prone position [27]. The use of an ARBP ensures that the simulation considers the complex anatomy of the breast, leading to more accurate and clinically relevant results. Furthermore, a more realistic HIFU ablation procedure was performed in this study, involving several lesion points at different areas of the tumour. The cumulative effect of these several lesion points on the fraction of necrotic tissue was evaluated, demonstrating the capability of this numerical platform for patient-specific simulation of real clinical HIFU therapies. Additionally, different abnormal tissue geometries are considered to investigate the effect of tumor types and shapes on HIFU ablation outcomes.

In this paper, we present a comprehensive approach to predict the outcome of the HIFU process using an ARBP. The simulation process involves several steps, starting with the generation of an acoustic wave and the calculation of the acoustic intensity distribution in the tissue phantom. The absorbed acoustic energy is then utilized as the heat source in the bioheat transfer model, allowing for the calculation of the amount of heat produced and the resulting increase in temperature in the tumour area. Finally, the fraction of necrotic tissue can be determined based on the temperature increase.

The paper is structured as follows: section 2 outlines the simulation procedure in detail, including the generation of the acoustic wave, calculation of the acoustic intensity distribution, and the bioheat transfer model. Section 3 presents the numerical results of the HIFU ablation, including the temperature increase and the fraction of necrotic tissue. Finally, section 4 concludes the study and summarizes the key findings.

## 2 Material and methods

The process of HIFU ablation is briefly described in this section. The controlling equations and mathematical models are then described. The subsequent subsections contain crucial information about the numerical modelling process at different steps.

## 2.1 HIFU ablation mechanism

HIFU ablation is a non-invasive technique where a transducer creates concentrated ultrasonic beams that penetrate soft tissue and focus on the targeted tumour [28]. Ultrasound beam focusing concentrates high intensities in a compact volume (e.g., 1 mm in diameter and 10 mm in length). In this method, the sound intensity induced in the focal region is about 100 to 10000 Wcm$^{-2}$, and peak compression pressures can reach 70 MPa, whereas peak rarefaction pressures can reach 20 MPa. Thermal effect and mechanical effect are two principal mechanisms involved in HIFU ablation [29].

The thermal impact of HIFU is heat creation because of the acoustic energy absorption with a fast rise in local tissue temperature. In most tissues, raising the temperature over 60°C for 1 second causes rapid and permanent cell death by coagulation necrosis, which is the principal mechanism for tumour cell elimination in HIFU treatment [29].

Mechanical effects caused by HIFU, such as cavitation, micro-streaming, and radiation force, are exclusively related to acoustic pulses at high intensities. Because the thermal process is better known and its effect is easier to regulate, it is favored in tissue ablation [29].

HIFU is quickly gaining clinical acceptability as a method for non-invasive ablation in various applications. Treatments usually require only one session, with the patient fully aware, gently sedated, or under light general anesthesia [30]. For a better understanding, the HIFU ablation process on a breast tumour is schematically illustrated in Fig 1.

To optimize hyperthermia treatment for breast cancer, it is essential to incorporate a patient-specific approach that begins with detailed modeling extracted from medical imaging (MRI/CT). These models help estimate the specific absorption rate (SAR) and understand the distribution of thermal energy necessary to achieve efficient tumor ablation while preserving adjacent healthy tissues. These concepts, including the illustration of different breast densities (from extremely dense to predominantly fatty), tumor depth and size, and the calculated SAR distribution, can be seen in the work by Singh et al. [31]. Such studies aids in comprehending how power is distributed and modulated to minimize the risk to surrounding healthy tissue—addressing the fringe heating phenomenon. A realistic simulation platform can be used for optimization of treatment procedure. The treatment can be tailored to manage the heat concentration precisely at the tumor location, thus sparing healthy tissues from unnecessary exposure. These realistic simulation studies not only enhance the efficacy of the treatment but also significantly reduce the collateral damage to the non-targeted areas, showcasing the pivotal role of computational modeling in personalizing cancer therapy. By extracting a patient-specific breast model from medical images, the purpose of this research is to achieve such simulation platforms mentioned that can be beneficial for pre-clinical purposes.

## 2.2 Governing equations

Here, we discuss the governing equations and procedures required to simulate the HIFU ablation process comprehensively. First, acoustic waves and their propagation inside the breast tissue are solved. Then, heat generation due to the absorption of these acoustic waves and heat transfer in the breast's biological tissue are simulated. Ultimately, the fraction of tumour damage and necrosis are predicted. The following are the governing equations in sequence for the solution procedure. s.

**Acoustic wave generation and propagation.** To model the stationary acoustic field, we solve the inhomogeneous Helmholtz equation. This equation is as [32]:

$$\nabla \cdot \left( -\frac{1}{\rho}(\nabla p - q_d) \right) - \frac{k^2 p}{\rho} = Q_m \tag{1}$$

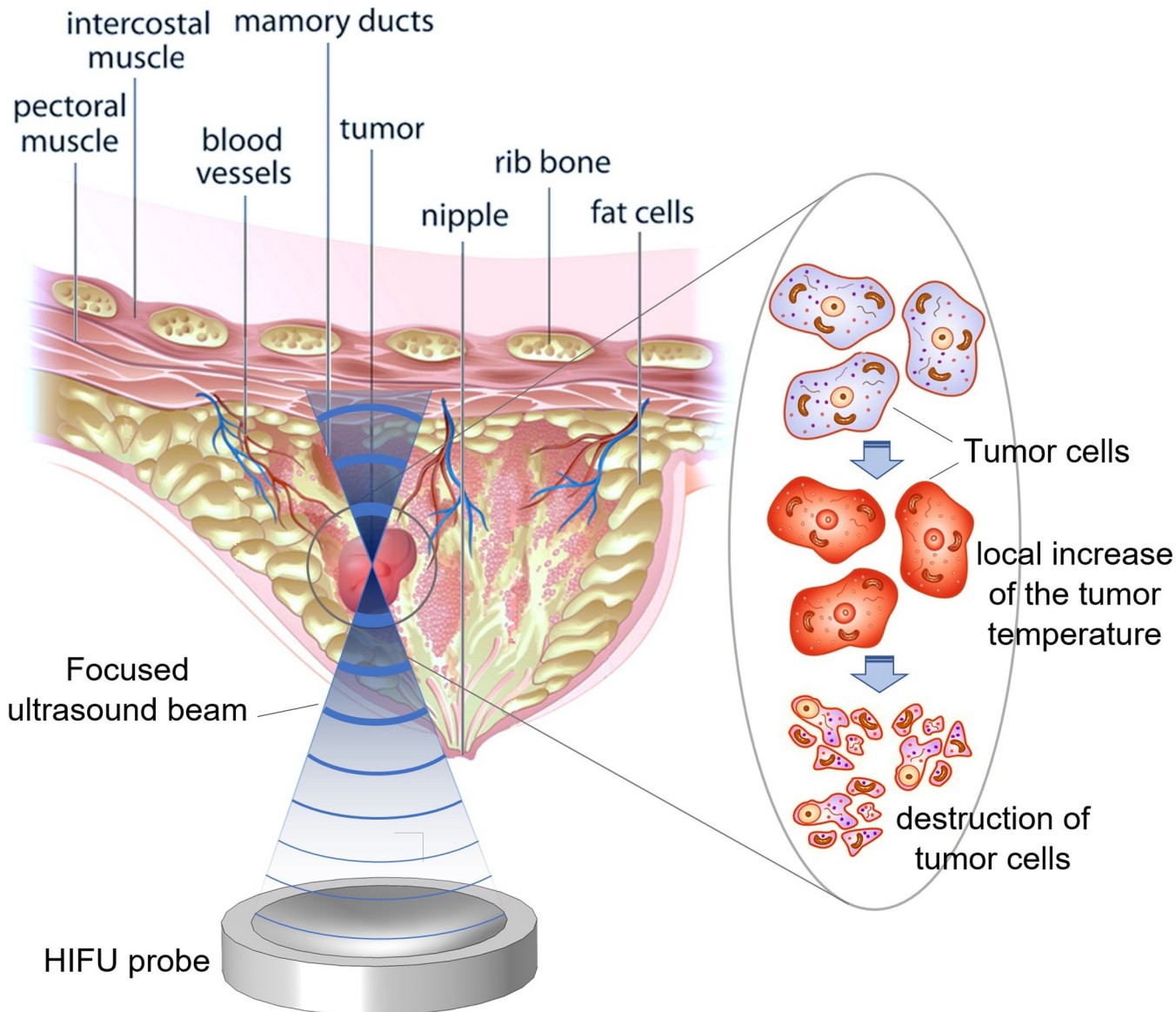

**Fig 1. A schematic of the HIFU ablation process for the treatment of a breast tumour.**

where $\rho$, $p$, $k$, $q_d$, and $Q_m$ are density, total pressure, wave number, dipole domain source, and monopole domain source respectively.

Acoustic waves will be attenuated as they propagate through breast tissues. Since various tissue types are considered, a user-defined attenuation model is applied for each tissue type with its specific attenuation coefficients ($\alpha_a$). Adding attenuation makes the wave number a complex value as in the following [32]:

$$k = \frac{\omega}{c} - i\alpha_a \qquad (2)$$

where $\omega$ and $c$ are angular frequency and speed of sound respectively.

**Bioheat transfer in breast tissues.** Here, Penne's bio-heat transfer equation is used to simulate temperature distribution in biological tissues [33]:

$$\rho c_p \frac{\partial T}{\partial t} = k_h \nabla^2 T + \omega_b \rho_b c_{p,b}(T_b - T) + Q_{met} + Q_S \qquad (3)$$

The above mathematical formula features several variables that relate to the physical characteristics of both tissue and blood. $\rho$ is the tissue's density, $c_p$ refers to the specific heat under constant pressure of the tissue, $k_h$ represents the heat conduction coefficient of the tissue, $\omega_b$ indicates the local blood perfusion rate, $\rho_b$ stands for blood density, $c_{p,b}$ symbolizes the specific heat of blood, $T_b$ refers to the local temperature of the arterial blood, $Q_{met}$ signifies the rate of metabolic heat generation in the area, and $Q_s$ denotes the heat source rate, which is equal to the power absorbed by the acoustic waves (the acoustic power that is dissipated).

When the acoustic field is solved, the acoustic pressure field and subsequently, the acoustic intensity field is computed. The $Q_s$ for thermal simulation is then calculated as [32, 34]:

$$Q_S = 2\alpha_a I = |Re(\frac{1}{2}pv)| \qquad (4)$$

In the above equation, $\alpha_a$ is the attenuation coefficient, $I$ is the acoustic intensity magnitude, $p$ is the acoustic pressure, and $v$ is the acoustic particle velocity vector. Also, sound intensity magnitude is the amount of power carried by the acoustic waves per unit area in a direction perpendicular to that area. It is defined as [35]:

$$I = p_A^2/2\rho c \qquad (5)$$

where $p_A$, $\rho$, and $c$ are the amplitude of pressure oscillation at the pressure anti-node, density, and speed of the sound respectively.

**Prediction of the fraction of necrotic tissue.** The degree of tissue injury due to the hyperthermia process can be evaluated by the Arrhenius kinetic model as in the following [36, 37]:

$$\frac{\partial \alpha}{\partial t} = (1 - \alpha)^n A e^{\frac{-\Delta E}{RT}} \qquad (6)$$

In the above equation, $n$ is the order of the polynomial in the equation, $R$ is the universal gas constant, and $T$ represents the tissue temperature. $A$ and $\Delta E$ are frequency factors and activation energy that are specific to the type of tissue and have been determined for various tissue types. In the case of breast tissue, these parameters were calculated to be $A$ = 1.18E + 44 $^{s-1}$ and $\Delta E$ = 3.02E + 5 Jmol$^{-1}$. After determining the degree of tissue injury ($\alpha$), the fraction of necrotic tissue ($\theta_d$)) can be expressed [37]:

$$\theta_d = \min(\max(\alpha, 0), 1) \qquad (7)$$

## 2.3 Geometry of the model

All simulations in this article are performed using the commercial finite element tool "COMSOL Multiphysics" [38]. An ARBP is used and placed inside a water tank. An immersed acoustic transducer is located at the bottom of the tank, producing an ultrasound acoustic wave that propagates inside the water and breast tissues. The transducer is bowl-shaped with a focal length of 53 mm chosen according to the location of the tumour's centre. The focal point of the transducer is considered to be at the centre of the tumour to produce a high amount of acoustic pressure in this region that leads to a sudden temperature increase and ablation of

tissues located in this region. This whole considered geometry is shown in Fig 2A. For a better demonstration of the acoustic transducer considered, a cross-section view of the model that passes through the middle of geometry is shown in Fig 2B. As can be seen in this figure, the transducer focal length (radius of curvature) and the transducer aperture are considered 53 mm and 60 mm respectively.

The numerical breast phantom used in the simulations was obtained from the UWCEM (the University of Wisconsin-Madison cross-disciplinary electromagnetics) repository of numerical breast phantoms [39], which also provided several other numerical breast phantoms with anatomical accuracy for detecting breast cancer that were derived from MRI scans. These phantoms were created using a set of MRIs taken from patients who were in a face-down position. The numerical breast phantom chosen, with Breast ID 012204 and classified as Class 2,

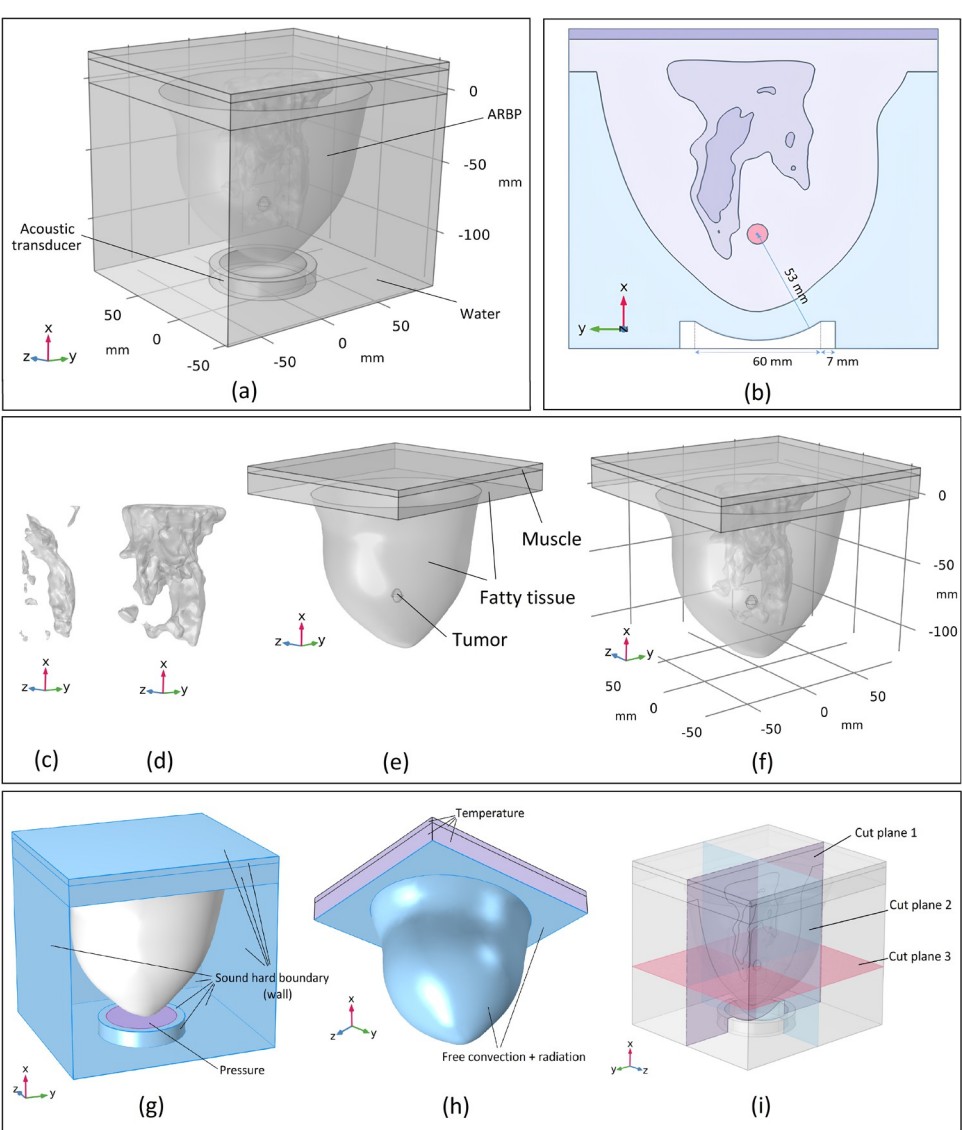

**Fig 2.** Computational domain details; (a) 3D visualization of the entire geometry considered; (b) middle cross-section view of the geometry; (c) FCG tissue; (d) transitional tissue; (e) fatty tissue, muscle, and tumour; (f) all tissues within each other as the ARBP; (g) boundary conditions considered in pressure acoustics physics; (h) boundary conditions considered in bioheat transfer physics; (i) considered cut planes that pass through the centre of the tumour.

was selected from the available options. According to the American College of Radiology, the following classification numbers correspond to their radiological density [40]; Class 1, mostly fat (less than 25% glandular tissue); Class 2, some fibro-connective/glandular tissue; Class 3, heterogeneously dense breast (51% to 75% glandular tissue); and Class 4, extremely dense breast (over 75% glandular tissue). The steps for bringing in and utilizing 3D grid-based numerical breast phantoms with anatomically accurate details in the COMSOL program can be find in our previous paper [8] in detail.

To create a simplified ARBP, four separate tissue types are extracted. Information pertinent to these tissue types is then collected from the provided database and utilized. Fibro-connective and glandular tissue (FCG), transitional tissue, fatty tissue, and muscle are the tissue types that were retrieved and taken into consideration. In our simulations, the tumour is modelled as a sphere with a 10 mm diameter. This kind of tumour, which is tiny and in the early stages of cancer growth, is categorized as T1c [41]. It should be noted that this tumour is situated inside the breast phantom's fatty tissue. All of the aforementioned tissue types are individually and together shown as the ARBP in Fig 2C–2F.

## 2.4 Numerical methodology and simulation procedure

In this section, the numerical modelling of HIFU is demonstrated and explained step by step. For each physics, the necessary data, material properties, and initial and boundary conditions are detailed. Initially, we demonstrate the simulation of ultrasonic wave generation within the domain. Subsequently, we conduct simulations of the heat generation and heat transfer within the breast phantom.

In the first step, "pressure acoustics, frequency domain" physics is considered, and an input pressure signal ($p_0$) is introduced at the transducer aperture boundary. The amplitude of this input signal is considered to be 0.1 MPa. According to Diaz et al. [42], the solutions reported for a similar HIFU apparatus of input below $p_0 = 0.35$ MPa remains in the linear regime. However, by increasing this input pressure, the results will go to a nonlinear regime and will deviate more and more from the linear solution, especially in the focal region. Here, solving the acoustic pressure equation (Eq 1) assumes that the propagation of acoustic waves is linear and that the magnitude of shear waves in the tissue is much less compared to that of the pressure waves. As a result, nonlinear effects and shear waves are not taken into consideration. Based on these assumptions, the input pressure equals 0.1 MPa. Also, the acoustic frequency was set to 1.5 MHz, which is a standard frequency for transducers used in HIFU therapeutic applications [43].

After the acoustic wave simulation and its propagation inside our breast phantom, the "bio-heat transfer, time-dependent" physics is added to be solved in the second step. The acoustic intensity field is calculated based on the acoustic pressure field. The heat source ($Q_s$) for thermal simulation is then calculated based on Eq 4. In our simulations, the acoustic waves are present for 10 seconds, and the ablation process takes 100 seconds. Therefore, the heat source generated due to the ultrasonic waves remains only for 10 s, and the simulation continues without this heat source for another 90 s to simulate the cooling down process and calculate the fraction of necrotic tissues due to this 100 s exposure of them to the generated high temperature. These exposure and therapy durations are similar to those in the Montienthong and Rattanadecho study [3] and can be further ratified by other clinical studies [44]. Thermal damage is added to the simulation to calculate the fraction of necrotic tissue, and the Arrhenius kinetics formula (Eq 5) is considered. The consideration of heat transfer through free convection and radiation to the environment is included in the simulation to produce accurate results. The effect of blood perfusion on cooling the breast and tumour area is also considered, as

detailed in Eq 3. The properties of blood, blood perfusion rate, and metabolic heat source for each tissue type are outlined in Table 1.

The simulation employs quadratic Lagrange discretization for both the pressure acoustics and bioheat transfer physics. The simulations were run on a system equipped with an Intel Core i5-8600 Processor with 3.10 GHz and 64 GB DDR4 RAM, taking approximately 127 hours to complete each simulation. Most of this computation time is for the simulation of the acoustic wave because of the very fine meshes needed for the simulation of this physics. The required mesh grids for each of the steps mentioned will be explained thoroughly in the following subsection. Various boundary conditions considered for each of these mentioned physics are shown in Fig 2G and 2H.

As seen in Fig 2G and 2H, for the "pressure acoustics, frequency domain", the boundary condition for the water tank and transducer's walls is considered a "sound hard boundary". For the chest's wall and other boundaries located inside the breast's tissue, since these boundaries are too far from the acoustic source (acoustic transducer's membrane), it can be an acceptable assumption to take them as "sound hard boundary" as well. "Sound hard boundary" is a boundary at which the normal component of the acceleration and, thus, the velocity is zero. For applying the input pressure signal, the "pressure" boundary condition is considered for the transducer's membrane, and as mentioned before, this input pressure is considered equal to 0.1 MPa. For the purpose of modelling heat transfer within the body, the "bioheat transfer" physics takes into account both heat flux and radiation from the skin to the surrounding water as boundary conditions for the skin. The temperature across internal boundaries, which are located inside the body, is assumed to be equal to the normal human body temperature of 37˚C and is set as the "Temperature" boundary condition for accuracy. The equations for these boundary conditions are provided [37]:

$$\text{Sound hard boundary (wall)} : -\vec{n} \cdot \left( -\frac{1}{\rho}(\nabla p - q_d) \right) = 0 \tag{8}$$

$$\text{Pressure} : p_0 = 0.1\text{E}+06 \tag{9}$$

$$\text{Convective heat flux} : -\vec{n} \cdot \vec{q} = h(T_\infty - T) \tag{10}$$

$$\text{Surface to ambient radiation} : -\vec{n} \cdot \vec{q} = \varepsilon \sigma_{SB}\left(T_\infty^4 - T^4\right) \tag{11}$$

$$\text{Temperature} : \ T = 310.15 \text{ K} \tag{12}$$

In the above equations, $\vec{n}, \rho, p, q_d, p_0, \vec{q}, h, T_\infty, T, \varepsilon$ and $\sigma_{SB}$ are surface normal vector, density, total pressure, dipole domain source, input pressure signal amplitude, heat transfer vector, heat convection coefficient, Surrounding water (ambient) temperature, temperature, surface emissivity, and Stefan-Boltzmann constant respectively. In the present study, the coefficient for heat convection from breast skin to the surrounding water is set at 50 Wm$^{-2}$K$^{-1}$. For human skin in water, heat transfer coefficients generally range from about 50 to 500 Wm$^{-2}$K$^{-1}$, depending on various factors such as water velocity, turbulence, and specific water conditions like temperature [45–48]. According to an experimental study carried out by Boutelier et al. [48], in still water, the heat transfer coefficients between human skin and cold still water were calculated to be 53.5 Wm$^{-2}$K$^{-1}$. Based on their experimental results and other research cited in their paper, we assume 50 Wm$^{-2}$K$^{-1}$ to be an appropriate value for this coefficient. The temperature of the surrounding water is assumed to be 293.15 K and the skin's surface emissivity is taken as 0.98 [49]. The properties of the materials used in the study can be found in Table 1.

**Table 1. Thermophysical and acoustic properties of different materials [1, 8, 45, 50–56].**

| Materials | $\rho$ (kgm$^{-3}$) | $C_p$ (Jkg$^{-1}$K$^{-1}$) | $k$ (Wm$^{-1}$K$^{-1}$) | $c$ (ms$^{-1}$) | $\alpha_a$ (dBm$^{-1}$MHz$^{-1}$) | $Q_{met}$ (Wm$^{-3}$) | $\omega_b$ (s$^{-1}$) |
|---|---|---|---|---|---|---|---|
| FCG tissue | 1050 | 3770 | 0.48 | 1470 | 87 | 700 | **0.0067** |
| Transitional tissue | 990 | 3270 | 0.345 | 1463.5 | 75 | 700 | **0.0067** |
| Fatty tissue | 930 | 2770 | 0.21 | 1457 | 48 | 700 | **0.0067** |
| Tumour | 1050 | 3852 | 0.54 | 1509 | 57 | 5790 | **0.005** |
| Muscle | 1100 | 3800 | 0.48 | 1588.4 | 109 | 5790 | **0.005** |
| Water | 997 | 4184 | 0.598 | 1480 | 0.22 | --- | --- |
| Blood | 1050 | 3617 | 0.52 | 1540 | 20 | --- | --- |

The table above includes several properties, including density ($\rho$), specific heat under constant pressure ($C_p$), heat conductivity ($k$), speed of sound ($c$), acoustic attenuation coefficient ($\alpha_a$), metabolic heat generation rate ($Q_{met}$), and blood perfusion rate ($\omega_b$). To evaluate the simulation results, three cross-sectional planes were considered, with intersections at the centre of the tumour, and are represented in Fig 2I. These three planes include the x-y, x-z, and y-z surfaces

## 2.5 Mesh study

The size of the mesh can be very important for the accurate simulation of the acoustic wave. For the simulation of the "pressure acoustics" physics, we need much finer meshes in comparison to the simulation of the "bioheat transfer" physics. So, two different mesh sets have been considered for solving these two physics.

**Pressure acoustics mesh.** For solving acoustic equations with high precision, a very fine mesh set is applied to the ARBP and the surrounding water with a maximum mesh size equal to λ/4 (where λ is the wavelength). The use of mesh sizes λ/4 for solving acoustic equations is mentioned in the context of high-precision finite element modeling [57]. Also, to accurately resolve the sharp pressure gradient in the focal region, the model uses an even finer mesh with a maximum mesh size of λ/8 for the tumour tissue. Guidelines from COMSOL Multiphysics emphasize using even smaller mesh sizes than λ/4 for reliable wave propagation simulations in high-intensity areas, indicating that finer meshes like λ/8 provide better accuracy [58]. So, the λ/8 mesh size is applied to the tumor area to resolve the sharp pressure gradient at this location, and an element growth rate of 1.2 is used, which gradually increases the mesh size as it moves further from the tumor. The mesh size becomes larger in regions farther from the tumor, as these regions have a more negligible effect on the calculation of correct results.

In this study, a number of computational meshes were created and the highest sound intensity magnitude at the focal point was calculated to evaluate the independence of the grid. The optimal number of grids was determined by finding a balance between computational cost and simulation accuracy. If the results of the simulation did not change with the use of finer meshes, the most refined grid was selected to achieve a suitable level of grid independence. The results of this mesh study for the "pressure acoustics" physics are shown in Fig 3A.

**Bioheat transfer mesh.** Much bigger meshes are required to solve bioheat transfer equations with high precision. So, to reduce the total simulation time, a different mesh set is considered for this physics. Similar to the previous, several samples of the computational mesh are also constructed for solving this physics. The maximum temperature generated at the focal point during the ablation process is measured for these samples to study the grid independency. Fig 3B shows the findings of this grid independency study for the "bioheat transfer" physics.

As can be seen in both Fig 3A and 3B, as the number of grids increases, the maximum intensity magnitude and the maximum temperature generated at the focal point saturate to a nearly constant level. Table 2 displays the numerical values and differences (inconsistencies

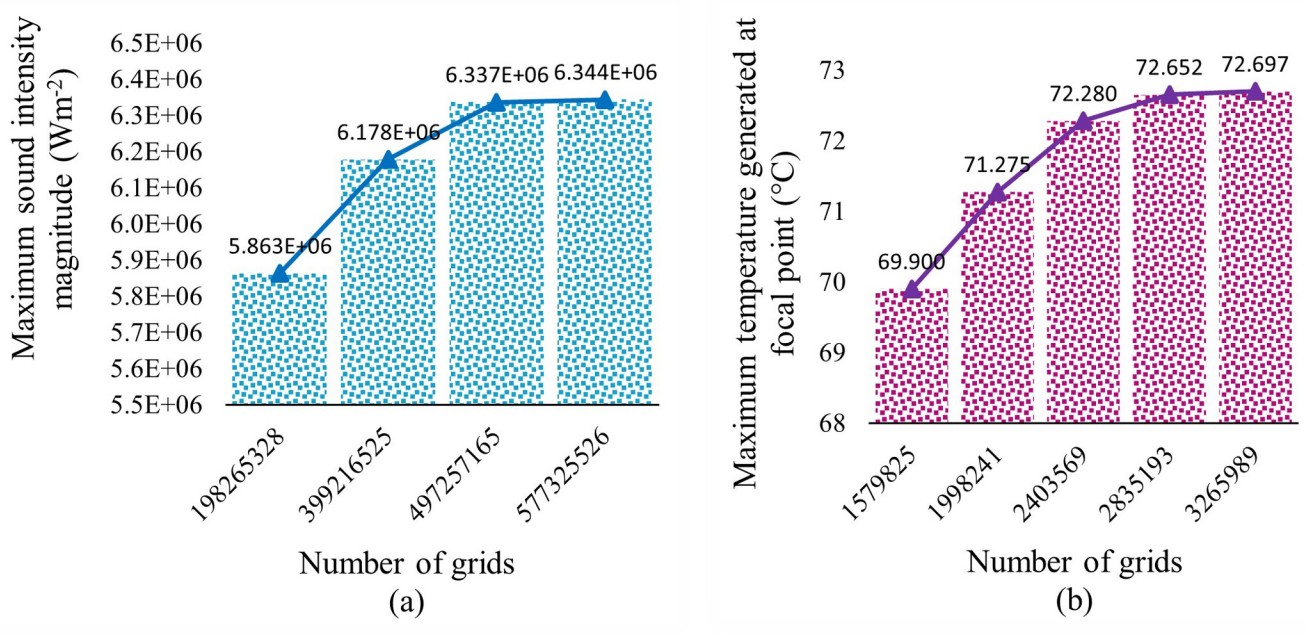

**Fig 3. Grid independency test for each physics.** (a) "pressure acoustics" physics, (b) "bioheat transfer" physics.

that arise between them) of multiple simulated scenarios considered with the objective of identifying the optimal scenario for both "pressure acoustics" and "bioheat transfer" physics. The maximum intensity magnitude for cases 1.3 and 1.4 are substantially equal, as shown in Fig 3A and Table 2. So, case 1.3 is selected as the optimum pressure acoustic mesh case for reducing computational expenses. Also, the maximum temperature generated at the focal point for cases 2.4 and 2.5 is approximately equal. So, for the "bioheat transfer" physics, case 2.4 is considered the optimum case.

The optimal mesh constructed for each of the "pressure acoustics" and "bioheat transfer" physics is shown in Fig 4A and 4C. To provide a clearer representation of each mesh grid and to view the generated mesh grids for each of the various tissue types, the meshed model is sliced and depicted in Fig 4B and 4D for each of these physics.

**Table 2. Quantitative amounts and relative errors of different considered mesh cases for each of the physics.**

| Pressure acoustic physics | | | |
|---|---|---|---|
| Case | Grid Numbers | Maximum sound intensity magnitude (K) | % Relative error in comparison with the optimum case |
| 1.1 | 198265328 | 5863212 | -7.471% |
| 1.2 | 399216525 | 6177986 | -2.504% |
| 1.3 | 497257165 | 6336653 | - - - - - - - |
| 1.4 | 577325526 | 6343974 | +0.116% |
| Bioheat transfer physics | | | |
| Case | Grid Numbers | Maximum temperature generated at focal point (K) | % Relative error in comparison with the optimum case |
| 2.1 | 1579825 | 69.89965 | -3.788% |
| 2.2 | 1998241 | 71.27549 | -1.895% |
| 2.3 | 2403569 | 72.27965 | -0.512% |
| 2.4 | 2835193 | 72.65199 | - - - - - - - |
| 2.5 | 3265989 | 72.69711 | +0.062% |

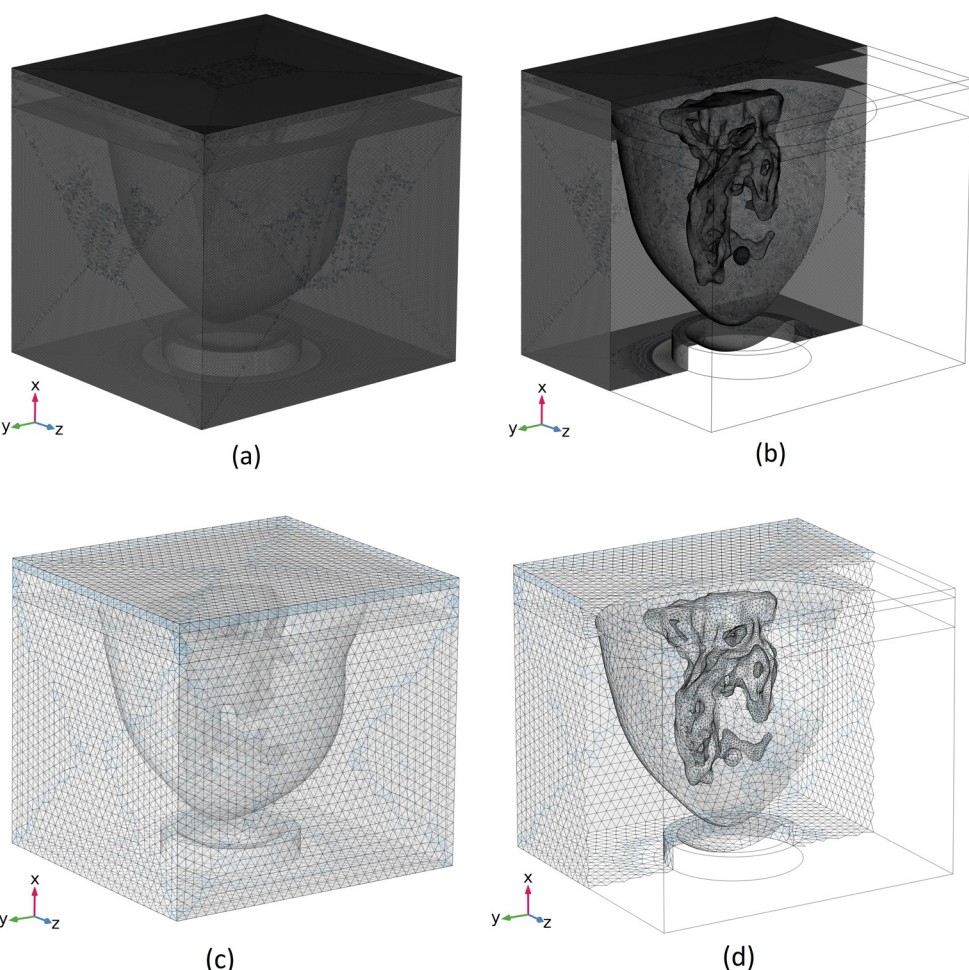

**Fig 4. Optimum grid generation considered for each physics.** "Pressure acoustics" physics: (a) whole geometry, (b) sliced magnified model. "Bioheat transfer" physics: (c) whole geometry, (d) sliced magnified model.

As mentioned before, the need for a highly dense mesh in the 3D simulation of HIFU in the FEM model is addressed by using two different sets of mesh grids. One set consists of very fine meshes for the simulation of acoustic waves, optimized separately by applying the necessary mesh sizes for high-precision acoustic simulation based on available guidelines and a comprehensive study balancing computational cost and simulation accuracy. The other set uses coarser meshes for the bioheat simulation, which does not require the very fine mesh used for acoustic waves. A similar comprehensive study was conducted for this mesh set to balance computational cost and simulation accuracy as well. By incorporating this approach, we were able to achieve a high-accuracy 3D simulation model of the entire real-size breast phantom.

## 2.6 Validation

To verify the accuracy of our simulation method, we conducted validation using two distinct previous studies. Initially, we validated our simulation procedure, by comparing our results with the simulation results of Montienthong and Rattanadecho's study [3]. Subsequently, we further examined the reliability of our simulation outcomes by comparing them with the simulation and experiment results obtained by Li et al. [59]. The purpose of conducting this second

validation process was to establish the reliability and precision of our simulation methodology when applied to real experiments. The details of these two validations are presented below.

**2.6.1 Montienthong & Rattanadecho simulation [3].** In this section for comparing our results with Montienthong & Rattanadecho ones, two steps are considered. which are simulations of the acoustic wave generation using the "pressure acoustics" physics, as well as the simulation of heat transfer using the "bioheat transfer" physics, These two steps are elaborated below.

*Pressure acoustics phenomenon.* For validating our first step of the simulation, Montienthong & Rattanadecho's results [3] are used. The study in [3] is a 2D axisymmetric simulation of HIFU in a cylindrical-shaped breast containing one sphere-shaped tumour and one vessel that passes through it. This cylindrical-shaped breast consists of three different tissue types. These tissue types are fat, glands, and muscle, and each is cylindrical-shaped. This breast tissue is considered inside a bigger cylinder as the surrounding water. The transducer is bowl-shaped with a hole in the centre and is immersed inside the water. They considered two different ultrasound frequencies equal to 1 and 1.5 MHz and two different tumour sizes equal to 5 and 10 mm. The same initial and boundary conditions as their study are adopted and the 1 MHz sample with the 5 mm tumour is simulated. Total acoustic pressure field and sound intensity magnitude distribution in breast tissue is obtained. For quantitative comparison of our results with theirs, the maximum total acoustic pressure and the maximum sound intensity magnitude in the whole domain are considered and compared. The plot of these quantities for both Montienthong & Rattanadecho's simulation and our simulation is shown in Fig 5A and 5B.

As evident, there are slight disparities between our simulation outcomes and Montienthong & Rattanadecho's simulation results. These little variations can be regarded as tolerable inaccuracy.

*Heat transfer phenomenon (Montienthong & Rattanadecho study [3]).* To validate the bioheat transfer prediction capability, we compare the present predictions with the results of Montienthong & Rattanadecho simulations. The same geometric setup and operation conditions are adopted. The predicted temperature distributions in the centre of the tumour (focal point) during the time-advancing ablation are presented in Fig 5C, together with Montienthong & Rattanadecho's results. As can be seen, the present study agrees well with the results of Montienthong & Rattanadecho's study.

A more in-depth comparative analysis is carried out using relative error, the results of which are displayed in Table 3. In this table, we compare the present simulations and the results of [3], concentrating on the maximum total acoustic pressure and the maximum sound intensity magnitude in the whole domain, and the maximum temperature obtained during the ablation process time at the focal point. A comparison of the results with those obtained by Montienthong & Rattanadecho using relative error analysis has revealed an acceptable level of error as shown in Table 3.

**2.6.2 Li et al.'s simulation and experiment [59].** To illustrate the disparity between our simulation results and the actual findings from experimental studies, Li et al.'s work is considered. They carried out several HIFU experiments and simulations and obtained the volume of heated necrosis element in an in-vitro bovine liver. Immediately following a session, the tissue was dissected to get the largest part of the coagulative necrotic volume, and the volume of the heated necrotic element was then measured. A heated necrotic element was defined as an area formed by a single HIFU treatment with a temperature more than 65˚C (temperature rise larger than 30˚C). They employed a therapeutic system made by Chongqing HIFU Technology Co., Ltd., and its transducer has the following dimensions: an inner radius of 4.0 cm, an outer radius of 7.5 cm, a radius of curvature of 9.3 cm, and a frequency of 1.6 MHz. In the current validation study, we considered their results obtained for 2 cm focal depth and acoustic

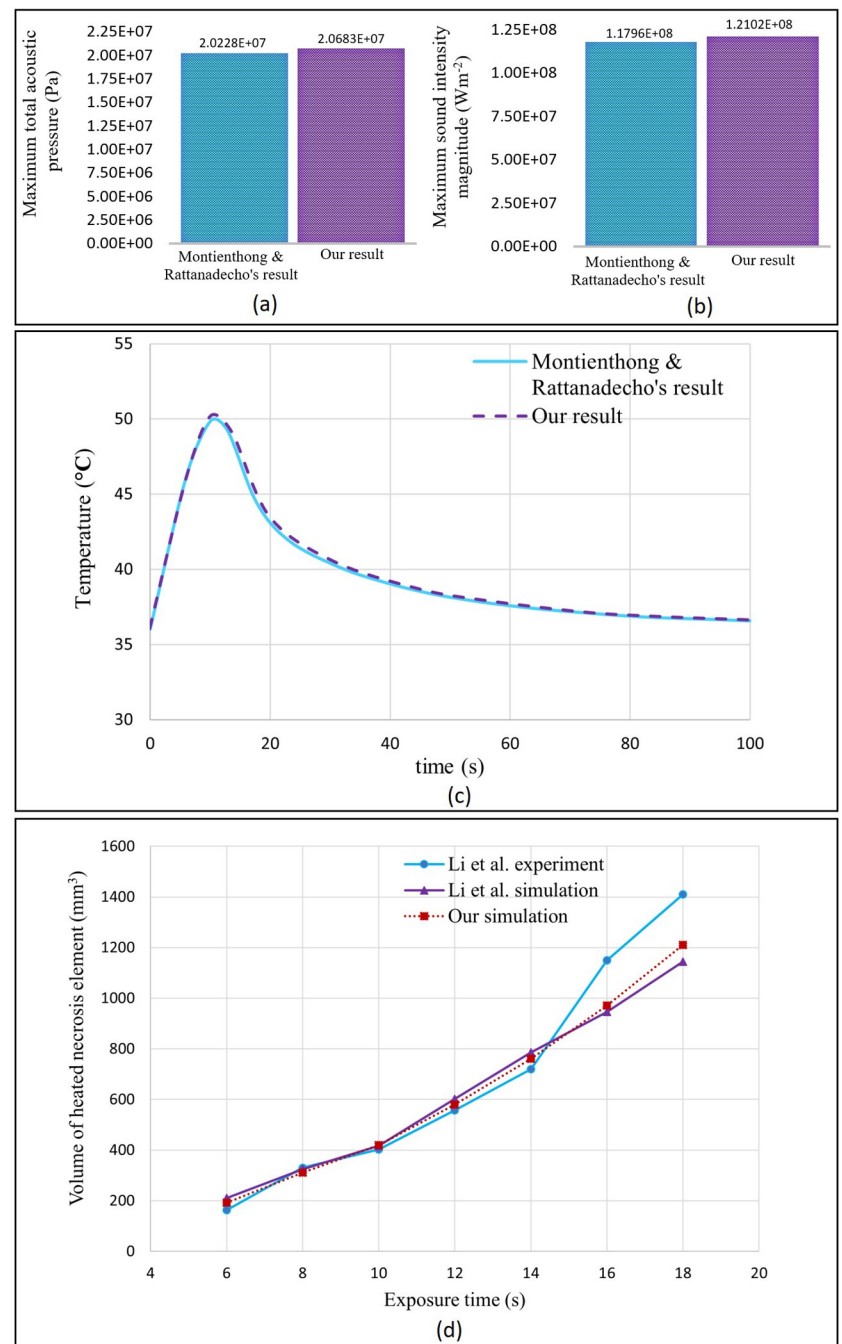

**Fig 5.** Validation study results; (a) comparison of the maximum total acoustic pressure obtained from our simulation and Montienthong & Rattanadecho's simulation; (b) comparison of the maximum sound intensity magnitude obtained from our simulation and Montienthong & Rattanadecho's simulation; (c) comparison of our simulation and Montienthong & Rattanadecho simulation plots of the temperature distribution in the focal point during the time; (d) comparison between Li et al.'s and our results for the volume of the heated necrotic element induced by HIFU at 2 cm focal depth and acoustic intensity of 25.4E+3 Wcm$^{-2}$ in an in-vitro bovine liver.

intensity equal to 25.4E+3 Wcm$^{-2}$. The same conditions are considered and simulated and the volume of the heated necrotic element is calculated. For the volume of the heated necrotic element caused by HIFU, various exposure durations of 6, 8, 10, 12, 14, and 16 s are taken into

**Table 3. Comparison between our simulation results and Montienthong & Rattanadecho's simulation results [3].**

| Parameters | Our result | Montienthong & Rattanadecho's result | % Relative error with Montienthong & Rattanadecho's result |
|---|---|---|---|
| Maximum total acoustic pressure (Pa) | 2.0683E+07 | 2.0228E+07 | +2.25% |
| Maximum sound intensity magnitude (Wm$^{-2}$) | 1.2102E+08 | 1.1796E+08 | +2.59% |
| Maximum temperature (°C) | 50.27 | 49.9 | +0.74% |

consideration. The plot of their experiment and numerical data, together with our simulation findings, is presented in Fig 5D.

As you can see in Fig 5D, our simulation results are almost the same as their results which indicates the veracity of our numerical method. But there is a gap between simulation results and experimental ones for samples with high HIFU exposure time. Similar to the observations made by Li et al. in their research, the observed disparities between the experimental and simulation results can be attributed to the omission of certain influential factors in the numerical simulation. These factors include nonlinear propagation of sound waves, cavitation formation, and alterations in the attenuation coefficient of tissue during exposure (coagulated tissue has different attenuation coefficient). Neglecting these effects, which become more prominent with increasing exposure time, contributes to the existing gap between the results obtained from experiments and simulations.

To address any concerns that may be raised about the methodology, it is important to emphasize the systematic approach adopted in our simulation studies. The computational methodology involved a rigorous mesh independence study, ensuring that the simulation results are not unduly influenced by the size or distribution of the mesh elements. This was achieved by progressively refining the mesh until a change in the solution was minimal, providing confidence in the stability and accuracy of our results. Additionally, a validation process was meticulously conducted against both prior studies and experimental data. For instance, we compared our simulation outcomes with those reported by Montienthong & Rattanadecho and further validated against experimental results from Li et al., enhancing the credibility of our simulation approach. Furthermore, the parameters used in the simulations, such as acoustic pressure, frequency, and tissue properties, were carefully selected based on published data and our previous findings to reflect realistic physiological conditions. By incorporating these rigorous steps, we ensure that our methodology is robust and that the results presented are both reliable and replicable. These systematic studies form the cornerstone of our computational approach, providing a detailed and scientifically sound basis for the simulation of HIFU ablation in breast tissue.

## 3. Results

This section illustrates results acquired from different simulated physics in different subsections. First, we show the total acoustic pressure field and distribution of sound intensity magnitude generated in the domain. Then, we present the induced temperature distribution in the breast model due to the absorption of this acoustic energy. Finally, the spread of necrotized tissue is displayed and we assess the quantity of tumour and breast tissue that undergoes necrosis due to the temperature rise. In the end, different tumor types and shapes are investigated and a more practical approach to HIFU ablation for breast tumors is explored, demonstrating the use of our numerical platform in predicting outcomes of HIFU ablation in actual clinical settings.

### 3.1 Acoustic field generation

As previously mentioned, a bowl-shaped acoustic transducer is used to generate a focused ultrasonic wave at a frequency of 1.5 MHz targeting the tumor area. The contours of total acoustic pressure and sound intensity magnitude are shown in Figs 6 and 7, respectively. The contours are shown for each of the cut planes 1 to 3 (three x-y, z-y, and x-z surfaces that pass through the middle of the tumour).

For a better illustration of the results relevant to the acoustics physics simulation, the plot of the acoustic pressure and intensity magnitude distribution along different x, y, and z directions are shown in Figs 8 and 9, respectively. These plots show the acoustic pressure and intensity magnitude distribution along lines that pass through the centre of the tumour in different directions. In these figures, the tumour location is shown by a hatched rectangle. It should be mentioned that the x-direction is the direction of the propagation of the acoustic wave.

After the "pressure acoustics" physics simulation, the "bioheat transfer" physics should be simulated to calculate the amount of heat and temperature increase due to this significant sound intensity generated in the tumour region. The results of this simulation are shown in the following subsection.

### 3.2 Temperature distribution

The high-intensity focused ultrasonic wave will generate heat in the focused area and increase the temperature in that region. The produced temperature distribution after 10 seconds on different cut planes along with the 3D view is shown in Fig 10. As mentioned before, we considered 100 s of total ablation time in which the pressure input pulse (the pressure field) is

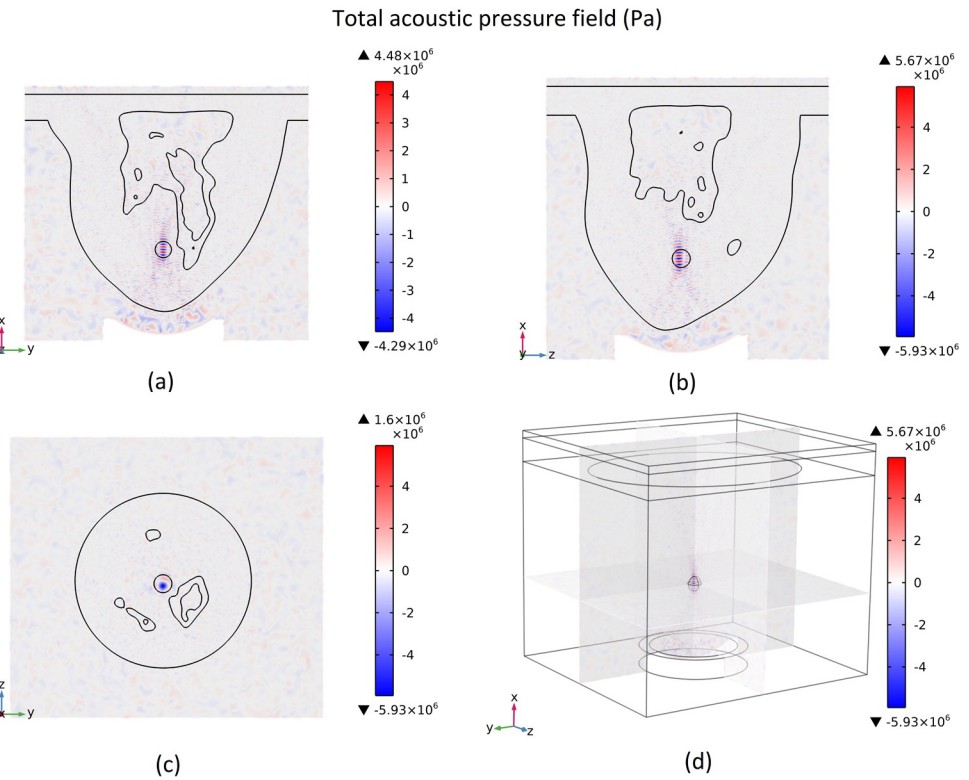

**Fig 6.** Total acoustic pressure field, (a) cut plane 1, (b) cut plane 2, (c) cut plane 3, (d). 3D view.

## Sound intensity magnitude (Wm⁻²)

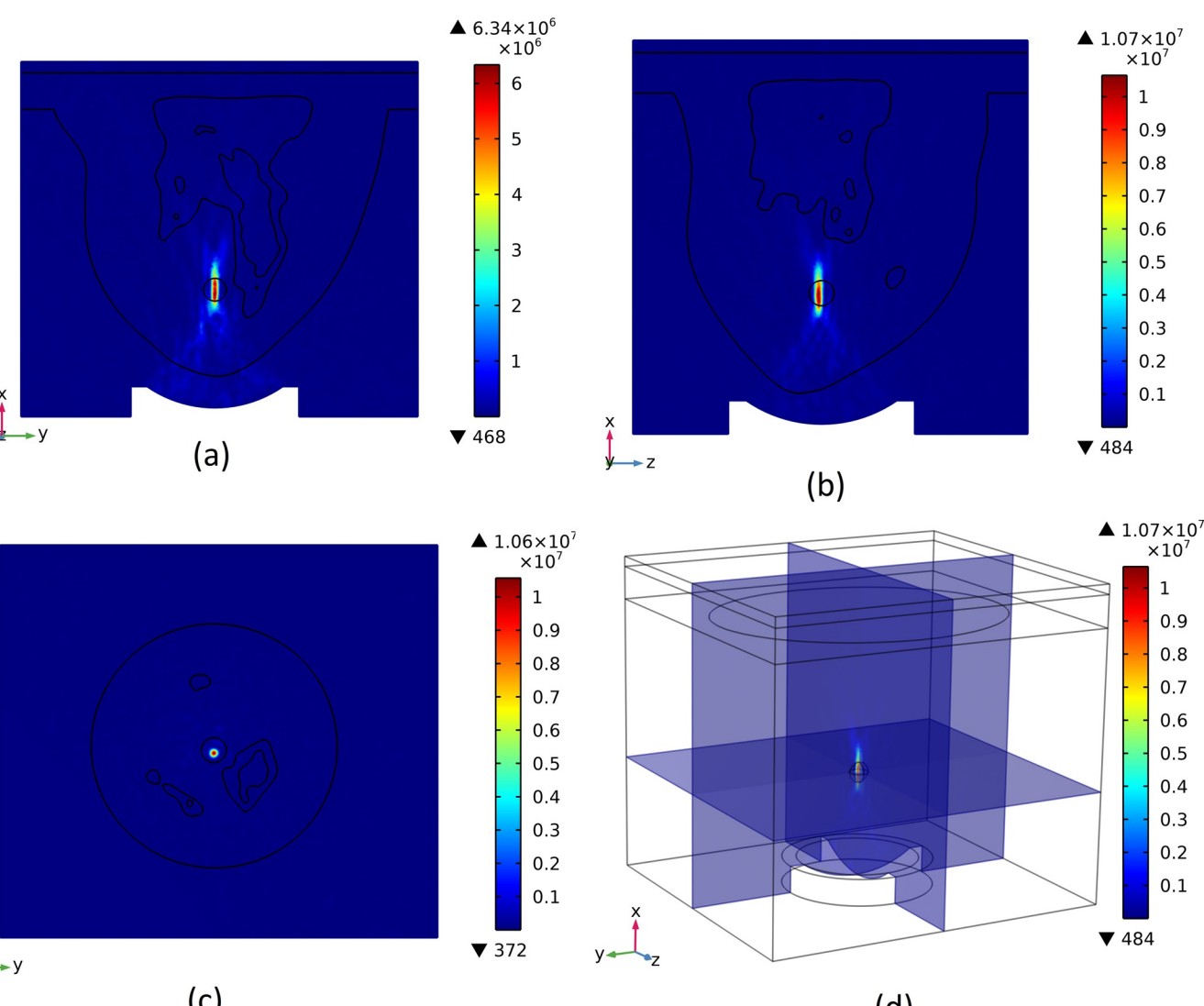

**Fig 7.** Sound intensity magnitude, (a) cut plane 1, (b) cut plane 2, (c) cut plane 3, (d). 3D view.

present for just the first 10 s. So, this figure shows the maximum temperature distribution achieved during the ablation process. Also, for a better illustration of the temperature rise in the tumour region, plots of the generated temperature after 10 s along lines passing through the tumour's centre in different x, y, and z directions are shown in Fig 11. A hatched rectangle shows the tumour region.

As mentioned before, the total ablation time is considered 100 s, and for calculating the fraction of tissues that necroses during this ablation time, the Arrhenius kinetic equation (Eq 6) is solved. The results of this part are shown in the following subsection.

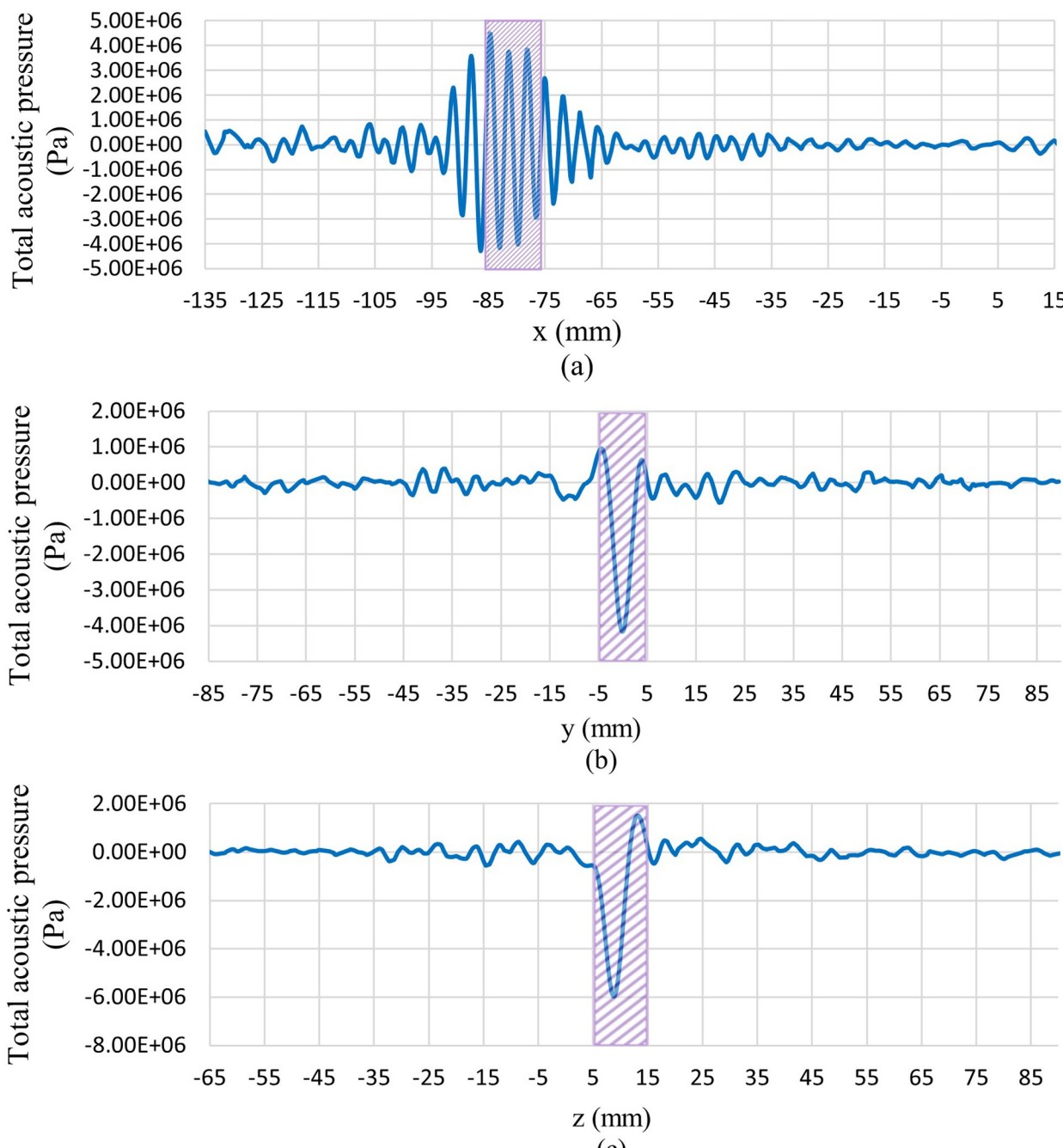

**Fig 8. Total acoustic pressure distribution along lines passes through the tumour's centre in different directions.** (a) x-direction, (b) y-direction, (c) z-direction.

### 3.3 Distribution of necrotized tissue fraction

The illustration of the distribution of necrotic tissue fraction after 100 seconds is presented in Fig 12. The contour of the fraction of necrotic tissue on various cut planes, as well as the three-dimensional view of the breast, is depicted in this figure.

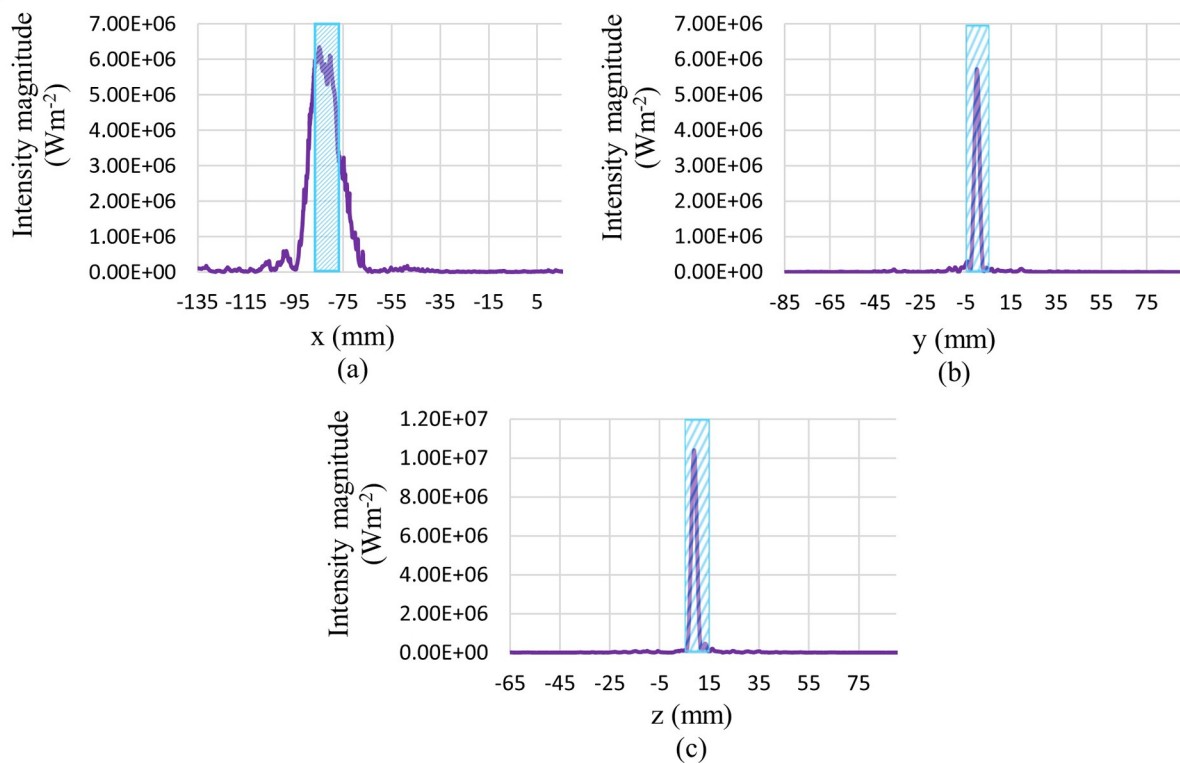

**Fig 9. Sound intensity magnitude distribution along lines passes through the tumour's centre in different directions.** (a) x-direction, (b) y-direction, (c) z-direction.

For a better understanding, plots of the distribution of the fraction of necrotic tissue after 100 s of the ablation process along lines passing through the tumour's centre in different x, y, and z directions are shown in Fig 13. The hatched rectangle represents the tumour location. Also, for a better illustration of the whole 100 s ablation process, plots of temperature and fraction of necrotic tissue at the centre of the tumour during the time are shown in Fig 14.

## 3.4 Realistic clinical approach

The previously mentioned approach which was just focusing the HIFU at the centre of the tumour for 10 s and then waiting for extra 90 s for thermal diffusion is not an efficient approach for a complete ablation of all the parts of a tumour (especially for relatively larger tumours) and is far from the nowadays clinical approaches. This mentioned process was only for showing different physics and steps involved in the HIFU phenomenon and to show the capabilities of our numerical platform. In this section, a more realistic clinical approach for HIFU ablation of breast tumours is considered and showed how our numerical platform can be utilized for the prediction of HIFU ablation outcomes in real clinical systems and different determined treatment procedures.

Nowadays clinical systems mostly operate by generating several lesions faster and covering the total volume of the tumour by moving the transducer focus to suppress uncertainties of the heat diffusion process and heat sink effect due to perfusion. So, here we considered a HIFU therapy process that includes 3 different faster lesions, one at the centre of the tumour and two others each at a 3 mm distance from the centre in the right and left sides of the tumour (in the y-axes). For this purpose, the considered transducer is moved accordingly and its focal point is

## Temperature after 10 s (°C)

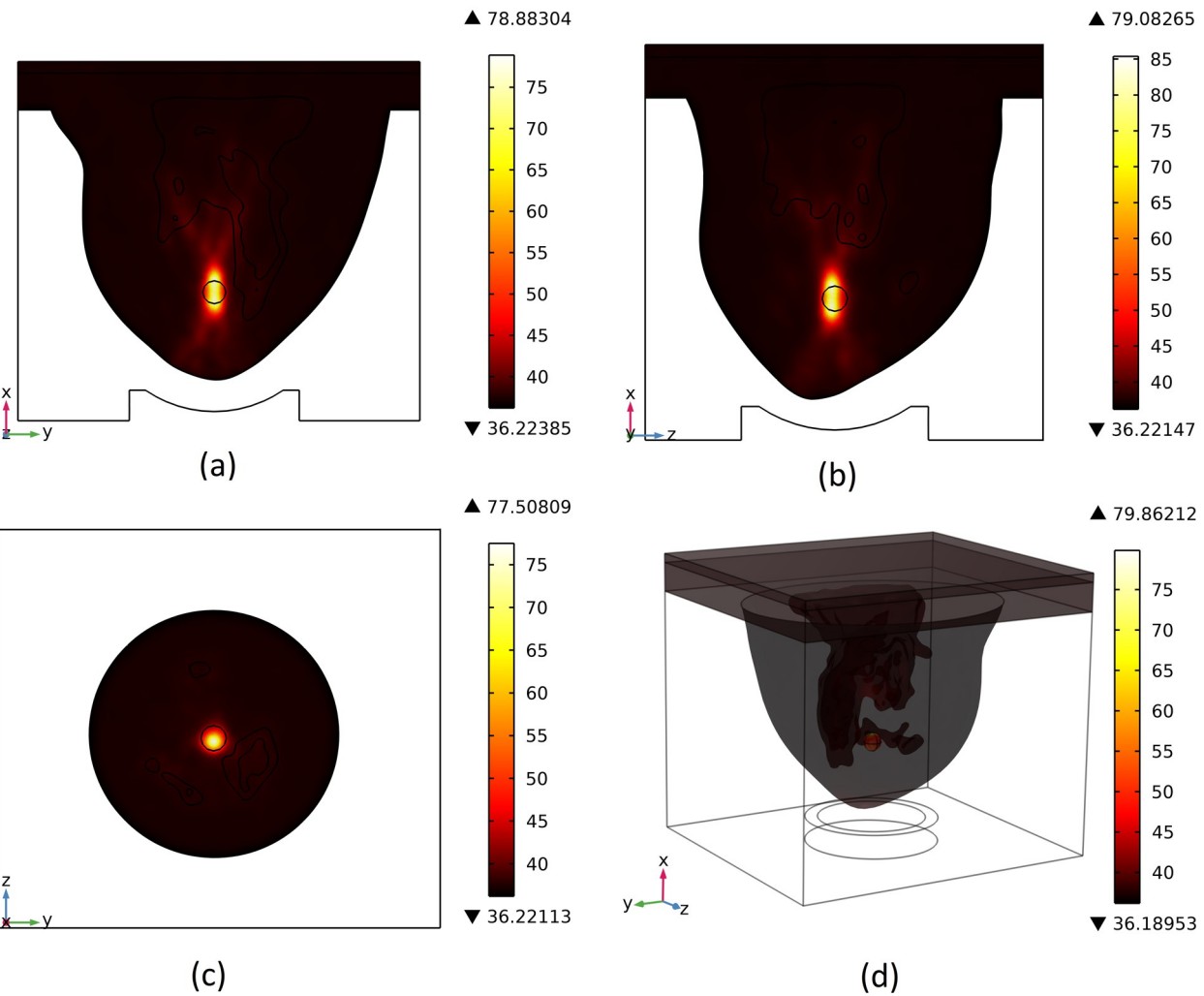

**Fig 10.** Distribution of temperature after 10 seconds, (a) cut plane 1, (b) cut plane 2, (c) cut plane 3, (d) 3D view.

adjusted at the mentioned points. So, first, the focal point is considered at the centre of the tumour and focused ultrasound waves are generated and kept for 6 s. 5 s is considered for moving the transducer and adjusting its focal point at a point 3 mm farther in the right side of the tumour. In this 5 s, no acoustic source existed and just the heat transfer and tumour necrosis are continuing. After this 11 s, focused ultrasound wave is generated on the right side for 6 s and then another 5 s without an acoustic source to move the transducer and focus it on a point with a 3 mm distance from the centre on the left side. Another 6 s is considered to heat this part of the tumour as well and then the simulation is continued without any acoustic source until reaching 100 s. This HIFU therapy procedure of generating multiple lesions and moving the transducer focus, along with the durations considered, is based on previous clinical approaches [60, 61].

The sound intensity magnitude distribution in the cut plane 1 (x-y surface that passes through the middle of the tumour) during each of these three lesion processes is shown in Fig

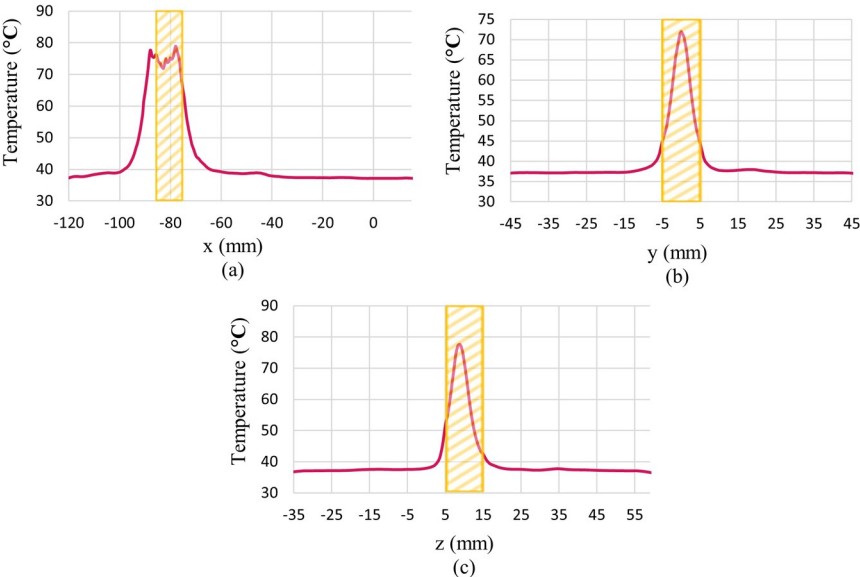

**Fig 11. Temperature distribution after 10 s along lines passes through the tumour's centre in different directions.**
(a) x-direction, (b) y-direction, (c) z-direction.

15A–15C. As mentioned before, the first lesion is from 0 to 6 s, the second from 11 to 17 s, and the third from 22 to 28 s. Also, the temperate and fraction of necrotic tissue distribution at the end of each mentioned lesion process in cut plane 1 is shown in Fig 15D–15I. The end of the first, second, and third lesions are at 6 s, 17 s, and 28 s, respectively. The fraction of necrotic tissue distribution after the whole ablation process (100 s) is shown in Fig 15J as well to show the final outcome of our HIFU ablation procedure.

For a better illustration of the results and to see how this more realistic clinical approach will affect the outcomes, plots of temperature and fraction of necrotic tissue along the lines passing through the tumour's centre in different x, y, and z directions are shown in Fig 16A–16F. These plots are illustrated at different times during the HIFU ablation process which is at the end of each lesion process and the end of the whole therapy. The tumour location is shown by a coloured rectangle in each of these plots.

## 3.5 Different abnormal tissue types

In this section, we investigated more realistic abnormal tissue shapes. Breast lesions are not always solid masses; they can also be non-mass enhancements or rim-enhancing masses. Therefore, comparing HIFU ablation results on these different lesion types can provide us with complementary information for further patient-specific HIFU simulations. Non-mass enhancements are areas of abnormal tissue in the breast that show enhancement on imaging but do not form a distinct lump [62]. Rim-enhancing masses are tumors with a bright ring around them on imaging, indicating a higher density of tissue at the edges [62].

We considered a geometry that replicates a small linear non-mass enhancement and another that replicates a rim-enhancing mass shape as the abnormal tissue in our ARBP. We performed the simulation procedure exactly as before and obtained results for each of these two cases with different abnormal tissue type. The geometries of these abnormal tissues, along with the relevant simulation results for these cases, including sound intensity magnitude, temperature after 10 seconds of HIFU exposure, and the fraction of necrotic tissue after 100 seconds of therapy, are illustrated in Fig 17.

## Fraction of necrotic tissue after 100 s

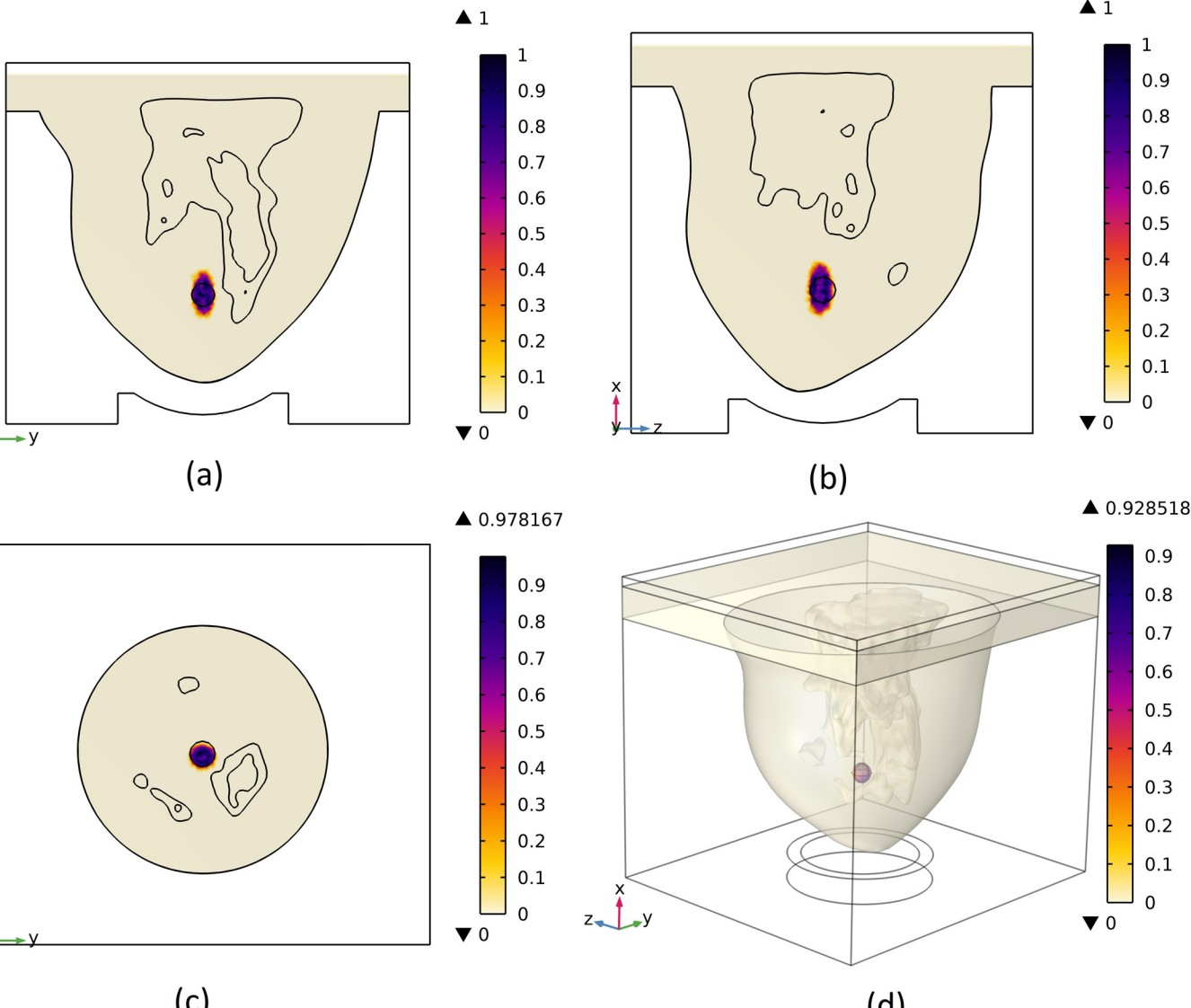

**Fig 12.** Distribution of necrotic tissue fraction after 100 s of the ablation process, (a) cut surface 1, (b) cut surface 2, (c) cut surface 3, (d) 3D view.

To better illustrate which parts of the tumor were ablated and which remained relatively undamaged, we used a threshold of 0.7 for the fraction of necrotic tissue and depicted the whole tumor, along with the undamaged portion, in Fig 18. This threshold means that if the necrotic tissue fraction in a region exceeds 70%, it is considered ablated. Conversely, regions with less than 70% necrosis are deemed nearly undamaged, suggesting the need for additional HIFU lesions to achieve full ablation.

The rationale for using the 70% threshold is based on two key factors. First, some cells in the ablation zone may survive initially but die later due to heat-induced damage, a process known as delayed necrosis. This occurs hours or days after treatment as cells fail to recover from structural damage. Second, even if cells do not die immediately, HIFU can cause sub-

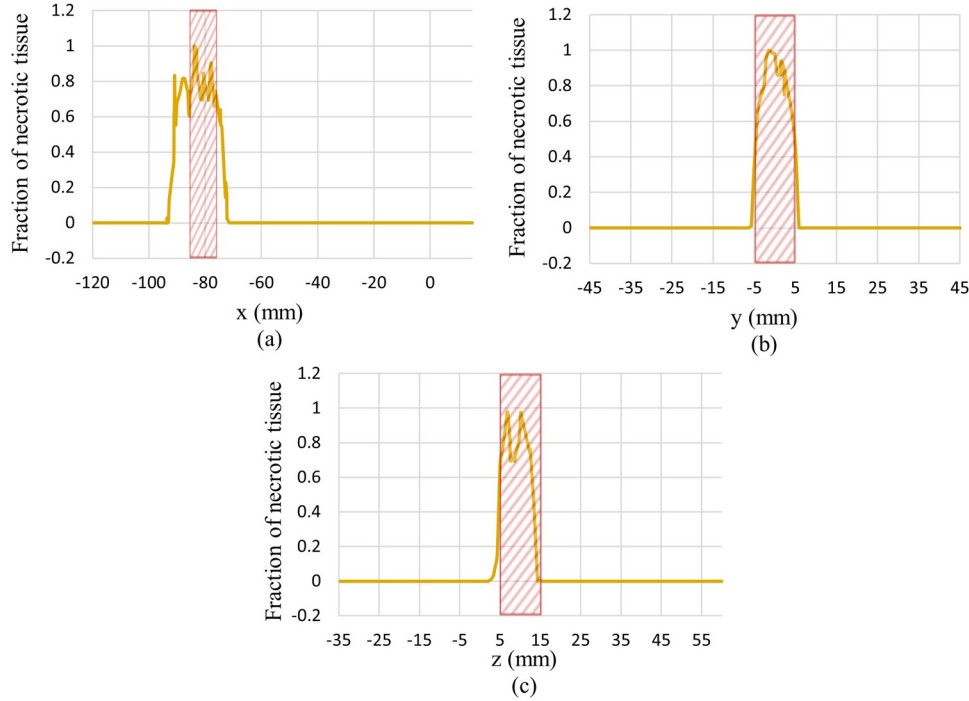

**Fig 13. Fraction of necrotic tissue after 100 s ablation process along lines pass through tumour's centre in different directions.** (a) x-direction, (b) y-direction, (c) z-direction.

lethal damage, impairing their ability to divide. These damaged cells may remain alive but lose the capacity for mitosis, preventing further tumor growth. Using the 70% threshold allows for effective tumor destruction while minimizing harm to surrounding healthy tissue.

In Fig 18, we present the necrotic tissue fraction for three different tumor types considered, as well as the corresponding volumes of tumor that remained relatively undamaged. This provides a clear illustration and qualitative analysis of which regions were successfully ablated and

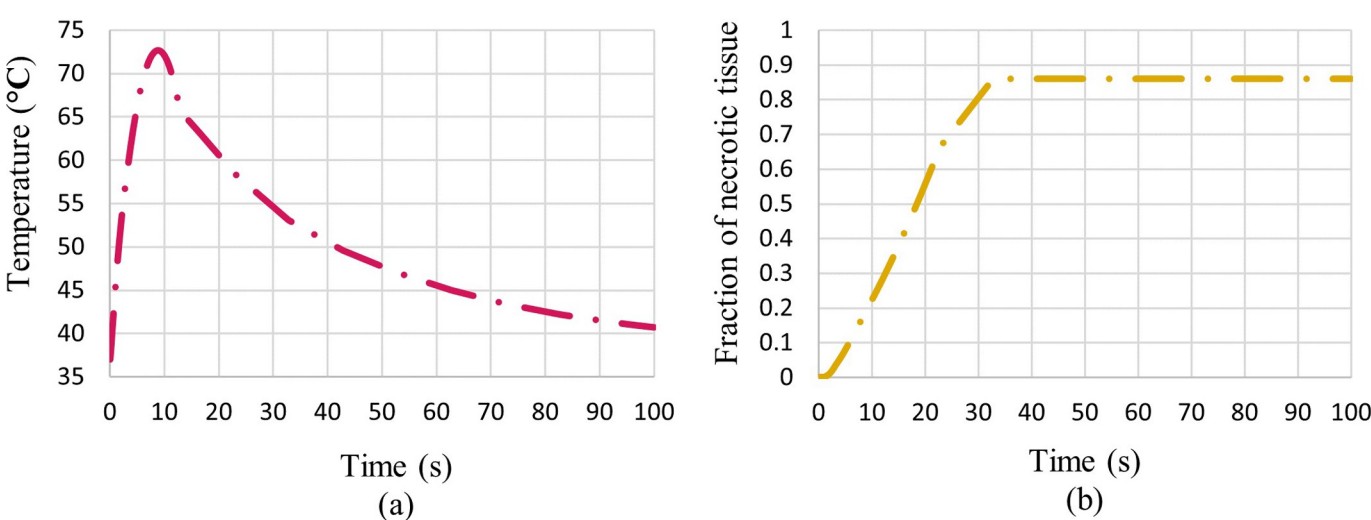

**Fig 14. Temperature and fraction of necrotic tissue at the tumour's centre during ablation time.** (a) Temperature change, (b) fraction of necrotic tissue.

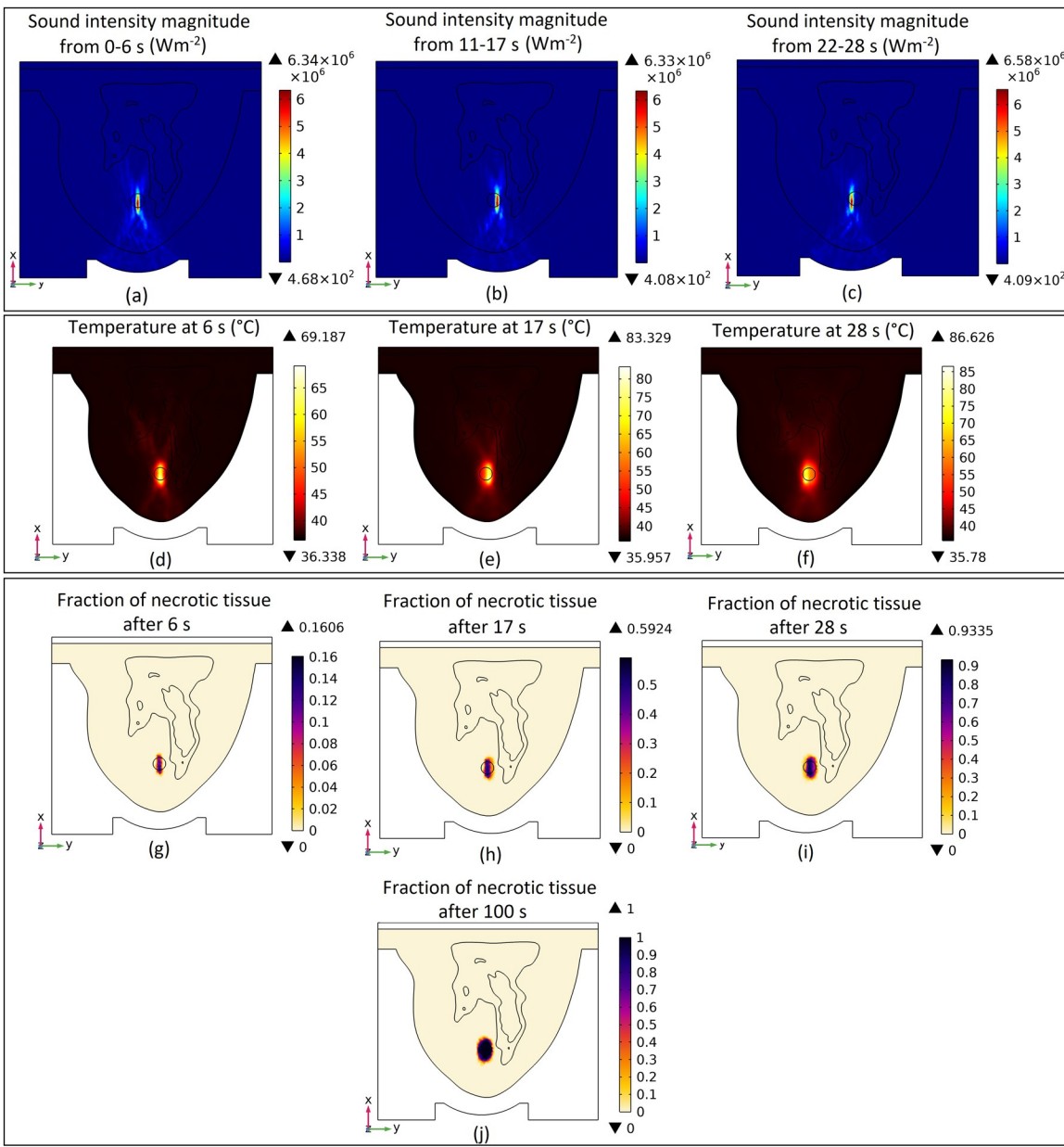

**Fig 15.** (a-c) sound intensity magnitude in the cut plane 1 during each of the three lesion processes, (d-f) temperate distribution in the cut plane 1 at the end of each lesion process, (g-j) fraction of necrotic tissue distribution in the cut plane 1 at the end of each lesion process and after the whole ablation process.

which areas may require further HIFU treatment for complete tumor ablation. Specifically, in the case of the solid mass, 32.33% of the tumor remained undamaged after one HIFU lesion. For the rim-enhancing mass, this value was 65.52%, and for the linear non-mass enhancement case, 79.68% of the tumor remained almost undamaged. These findings emphasize the importance of lesion planning, especially for complex tumor structures such as rim-enhancing and non-mass enhancing lesions, where a single ablation may leave a substantial portion of the tumor untreated.

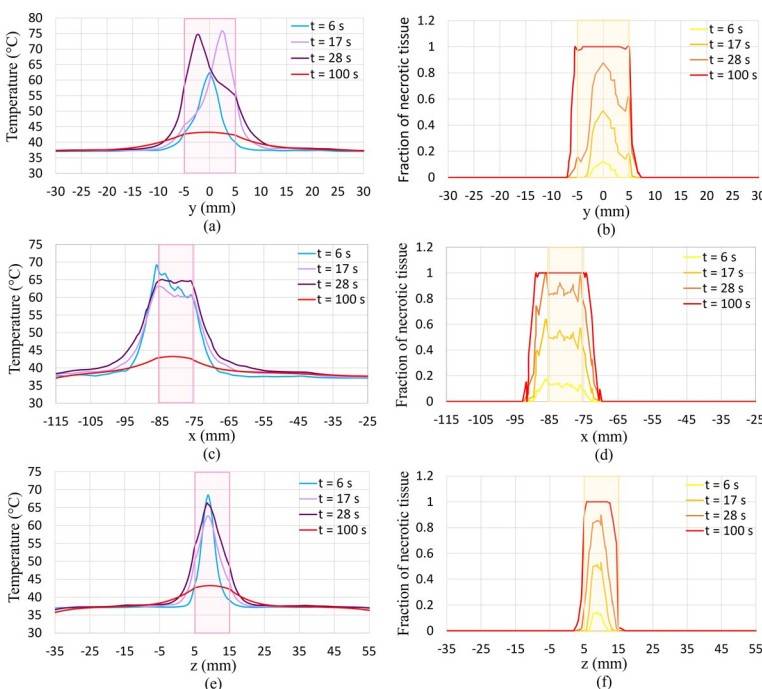

**Fig 16. Temperature and the fraction of necrotic tissue after 6 s, 17 s, 28 s, and 100 s along lines pass through the tumour's centre in different directions.** (a and b) y-direction, (c and d) x-direction, (e and f) z-direction.

## 4 Discussion

In this section, we will provide a detailed discussion of each of the results mentioned, identify the limitations and drawbacks of our simulation platform, explain the reasons for these limitations, and propose potential methods to overcome them in possible future studies.

As can be seen in Figs 6 and 7, the acoustic wave is focused at the centre of the tumour. This focused ultrasonic wave will produce a large amount of energy in the tumour area that will be converted to heat, increasing the tissue's local temperature and leading to tumour ablation. As it is shown in Fig 6, the amplitude of acoustic pressure is much higher in the tumour area, and it reaches +5.67 and -5.93 MPa at its highest point. This high amount of total acoustic pressure means a higher sound intensity magnitude. In Fig 7, it is shown that the sound intensity magnitude can reach 1.07E+07 Wm$^{-2}$ at its highest point, which is the centre of the tumour.

As shown in Fig 8, the amplitude of pressure oscillations is much higher at the tumour location due to the focusing of acoustic waves in this region. This higher pressure amplitude leads to a higher amount of sound intensity, which means more energy that the acoustic waves carry in this region. As shown in Fig 9, we see a large peak in the sound intensity magnitude in the tumour region. This leads to a sudden generation of heat in this region due to the dissipation of this high acoustic power.

By comparing Fig 10 with Fig 8, the relation between the amount of the temperature rise and the sound intensity magnitude can be seen. Obviously, in areas with higher sound intensity magnitudes, the temperature increase will be higher after 10 s exposure. As shown in Fig 10, the temperature rose in the tumour area, and the maximum temperature achieved was about 80˚C. Also, as Fig 11 shows, due to tumour exposure to a high-intensity focused ultrasonic wave for 10 s, a prominent peak of temperature distribution occurred in the tumour region, and the temperature at the centre of the tumour rose to approximately 80˚C. This

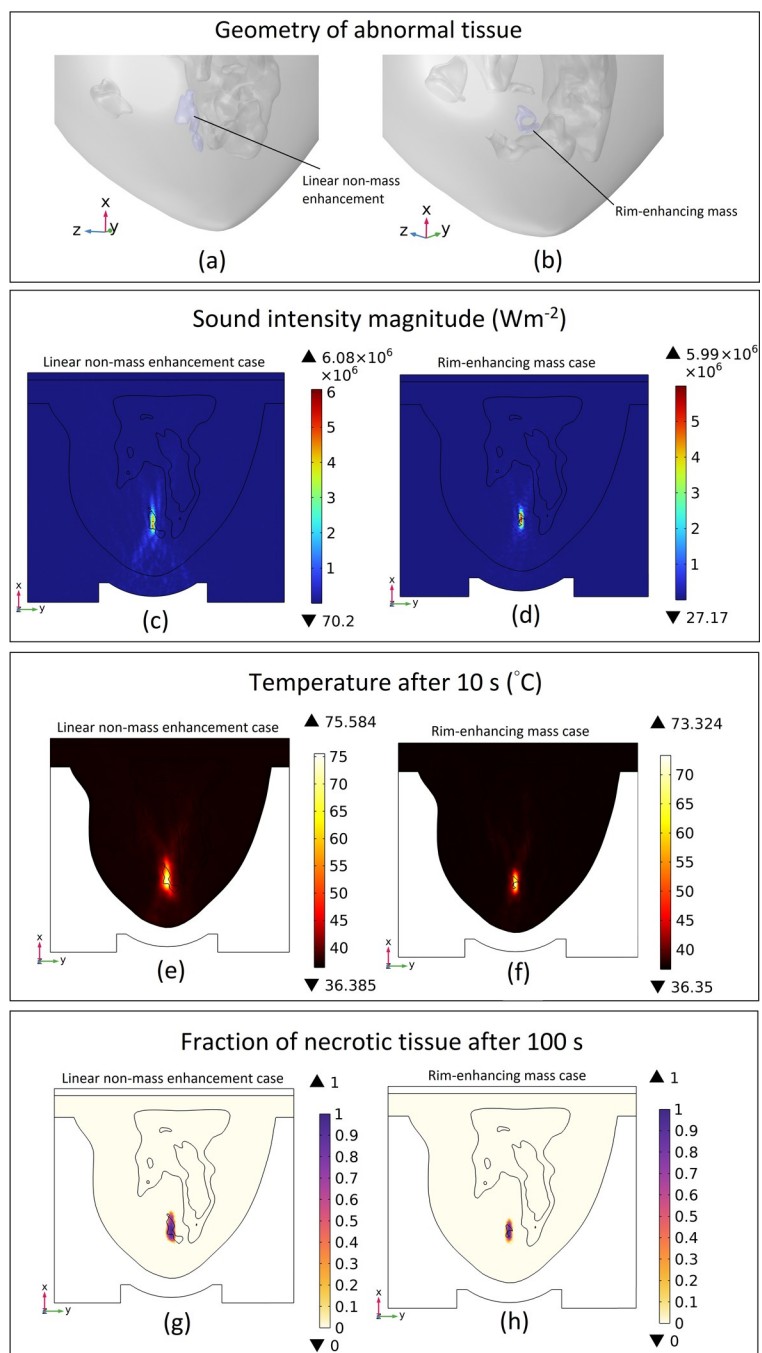

**Fig 17.** (a & b) Geometries of abnormal tissues: linear non-mass enhancement and rim-enhancing mass, respectively, (c & d) sound intensity magnitude in the cut plane 1 for each of the cases, (e & f) temperate distribution in the cut plane 1 after 10 seconds for each of the cases, (g & h) fraction of necrotic tissue distribution in the cut plane 1 after 100 s ablation process for each of the cases.

extreme temperature can cause a sudden killing of tissues. Also, temperatures between 46 to 56°C are considered thermal ablation ranges. These temperatures can cause tissues to undergo coagulation or tissue necrosis, called thermo-ablation if the tissue is kept in these temperatures long enough [63].

# Fraction of necrotic tissue in tumor after 100 s

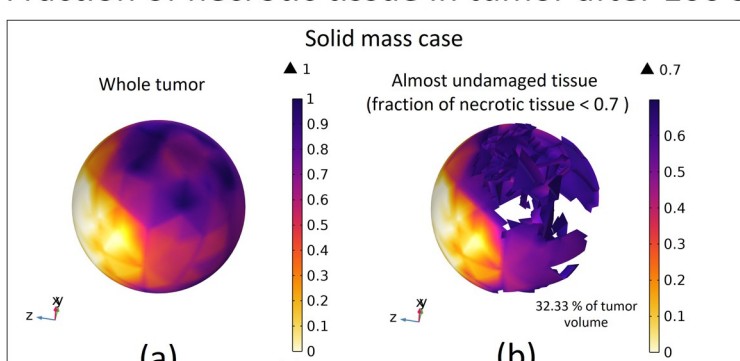

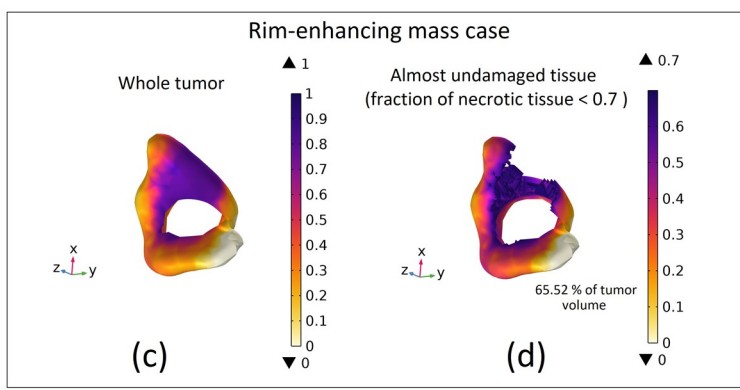

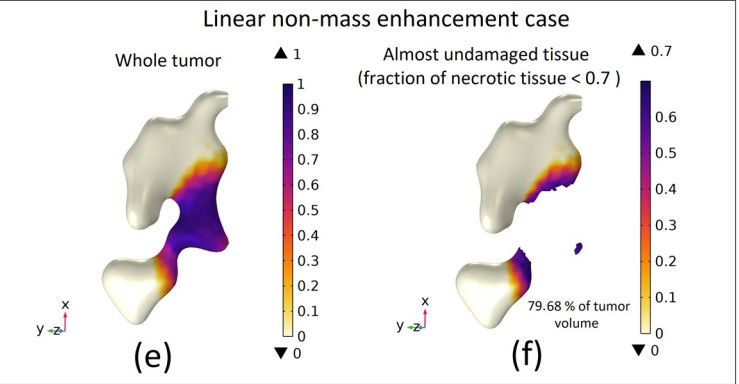

**Fig 18. 3D distribution of the fraction of necrotic tissue after 100 s of ablation for different tumor types.** (a & b) whole tumor geometry and almost undamaged tissue (fraction of necrotic tissue < 0.7) for the solid mass case. (c & d) Whole tumor geometry and almost undamaged tissue for the rim-enhancing mass case. (e & f) Whole tumor geometry and almost undamaged tissue for the linear non-mass enhancement case.

As demonstrated in Fig 12, almost the entire tumour has been destroyed after 100 seconds, and a minor portion of the normal tissue located near the tumour has also perished. The highest amount of necrotic tissue fraction achieved is one which means all portion of tissue in that area has been necrosed. As illustrated in Fig 12 and compared to Fig 10, it is evident that there is a correlation between the extent of necrotic tissue fraction and the increase in temperature. In regions where the temperature rises to higher levels, the tissue ablated after 100 seconds of exposure will also be greater.

As seen in Fig 13, the fraction of necrotized tissue at the centre of the tumour reaches one, but this amount will decrease gradually by going farther from the tumour's centre. As can be seen, not all parts of the tumour are ablated completely. For the complete ablation of all tumourous tissue, by altering different parameters, several procedures can be applied by a clinician. First, by briefly dislocating the acoustic pulse generator, the focal point will be changed. So, the clinician can ablate different locations of tumourous tissue more efficiently. Second, by increasing the pressure of the input pulse, they can reach a higher ablation temperature at the focal point which leads to a higher fraction of tissues that will be necrotized. Third, by increasing the input acoustic pulse duration, they can expose the tumour to ablation temperature for a longer time which leads to the ablation of a higher fraction of tumourous tissue. Alternatively, a clinician can consider several sessions of HIFU ablation therapies to ablate parts of the tumour that were not ablated in just one session.

As can be seen in Fig 14A, for the first 10 s that the acoustic pressure field exists, the temperature rises sharply in the tumour's centre from 37˚C to around 75˚C. After that, by eliminating the HIFU source, the tumour's centre's temperature gradually reduces to 37˚C due to the heat transfer and the cooling effect of the blood perfusion process. On the other hand, as shown in Fig 14B, the necrotic tissue fraction at the tumour's centre rises continuously to around 0.9 after 100 s. It means that 90% of tumourous tissues are ablated at the tumour's centre.

For the realistic clinical approach scenario, as can be seen in Fig 15, three different focal points are considered in order to apply three different lesions consecutively and a sound intensity magnitude of around 6E+6 Wm$^{-2}$ is produced at each of these focal points. These high sound intensity magnitudes will cause the temperature to increase to the highest amount of around 69, 83, and 86˚C at the end of each lesion process (at different considered lesion points in the tumour) which will lead to more uniform tumour necrosis. As shown in this figure, the fraction of necrotic tissue is equal to 1 at almost all parts of the tumour after 100 s HIFU therapy process which means all parts of the tumour are ablated.

The temperature rise at the centre, right and left sides of the tumour (at different considered lesion points) can be seen in Fig 16A which produce a more uniform high-temperature distribution at different parts of the tumour. If you compare Fig 16B, 16D and 16F with Fig 13, you can see in this more realistic clinical procedure, the fraction of necrotic tissue at almost all parts of the tumour in every different direction reaches 1. While this amount is not 1 at the verges of the tumour in the previous approach which was just one longer lesion at the centre of the tumour. This means this approach is more effective in killing tumour tissues uniformly and completely. By increasing the number of lesion points in various parts of the tumour, more desirable results can be obtained and also, we don't need a long lesion time. This is what is happening in current clinical approaches. This numerical platform has the capability to be used for simulating any HIFU ablation process with various characteristics such as different lesion numbers, duration, locations, different acoustic generation power, frequency, etc. This tool can be very useful for predicting different HIFU ablation therapy outcomes and can be utilized to choose the most efficient procedure based on the size and location of the tumour.

As depicted in Fig 17, our simulation platform enables the prediction of HIFU procedure outcomes for various types of abnormal tissue. Comparing Fig 17C and 17D with Fig 7A reveals that the sound intensity magnitude at the abnormal tissue area is higher for solid mass tumors compared to non-mass enhancements, approximately 6.34E+06 Wm$^{-2}$ for the solid sphere tumor versus 6.08E+06 Wm$^{-2}$ for the non-mass enhancement case. Solid mass tumors have a more concentrated and uniform structure, facilitating better transmission and absorption of ultrasound energy, resulting in higher sound intensity within the tumor tissue. In contrast, non-mass enhancements, with their less uniform and more diffuse structure, scatter ultrasound energy more, leading to lower sound intensity at the abnormal tissue site.

Furthermore, comparing non-mass enhancement with rim-enhancing mass, we observe lower sound intensity magnitude at the abnormal tissue area in rim-enhancing mass. approximately 5.99E+06 Wm$^{-2}$ for the rim-enhancing mass versus 6.08E+06 Wm$^{-2}$ for the non-mass enhancement case. This could be attributed to the presence of a cavity in the rim-enhancing mass, where only the rim constitutes abnormal tissue, while the most of area consists of normal fatty tissue, which transmit less energy and thus reduces the overall sound intensity at the abnormal tissue area.

Consequently, for the same reason, comparing Fig 17E and 17F with Fig 10A, shows that temperature increase is highest in solid mass tumors (around 79˚C), followed by non-mass enhancements (around 76˚C), and lowest in rim-enhancing masses (around 73˚C). This temperature variation correlates with the extent of necrotic tissue formation, where solid mass tumors exhibit the largest necrotic area, followed by non-mass enhancements and rim-enhancing masses.

By analyzing Fig 18, we can observe the extent of tumor ablation and the regions that remained nearly undamaged for each tumor type. For the solid mass tumor, a single lesion successfully ablates a significant portion of the tumor volume (67.67%), with 32.33% of the tumor, primarily near the boundary, remaining undamaged. To achieve complete ablation, additional lesions are necessary, focusing on these boundary areas. This aligns with the procedure outlined in the previously mentioned "Realistic Clinical Approach" section, where multiple lesion points are used to ensure comprehensive coverage of the tumor.

In contrast, for the rim-enhancing mass case, the majority of the tumor remains undamaged (65.52%), with only 34.48% ablated. This is due to the ring-shaped geometry of the tumor and the focal point of the transducer being placed at one specific location. Since the tumor has a non-uniform shape, a single lesion cannot influence most of the tumor, and thus, several focal points are required to encompass the entire tumor region. The figure provides a qualitative analysis of the ablation process, helping to identify the regions that require further treatment. This insight is crucial for developing an optimized treatment trajectory, ensuring that future lesions target areas that have remained undamaged.

For the linear non-mass enhancement case, only 20.32% of the tumor is ablated, while 79.68% remains undamaged. This can be attributed to the large volume and diffuse nature of the tumor, which lacks a concentrated mass. As a result, a single lesion is insufficient for complete ablation. This figure offers a spatial representation of the tumor, showing the regions that should be targeted for the next set of lesions, and helps guide the trajectory of the transducer applicator for an optimized treatment strategy.

Addressing treatment planning challenges, particularly in cases with rim-enhancement and non-mass enhancements, is crucial. The uneven shape of these tumors complicates the ablation process, as the transducer focal point may not fully encompass the tumor, leading to incomplete coverage. These cases require a careful and adaptive approach to ensure that peripheral regions, which are often difficult to treat, are adequately targeted. Our simulation platform is a powerful tool for addressing these challenges, particularly in cases with complex tumor geometries such as rim-enhancing and non-mass enhancement tumors. By providing a 3D representation of the tumor and surrounding tissue, distinguishing between ablated and undamaged areas, the platform allows for precise targeting of undamaged regions and helps determine the optimal transducer trajectory. This enables clinicians to strategically plan multiple lesion points, ensuring that peripheral regions of the tumor are effectively treated, which is essential for achieving complete ablation. Furthermore, the platform's ability to evaluate the number of lesions, their duration, and power settings makes it highly adaptable to patient-specific needs, thus improving the effectiveness of HIFU therapy and reducing the risk of incomplete treatment.

It is worth noting that 3D simulations provide much more accurate results compared to 2D simulations tackled by previous researchers. In 3D models, due to the added complexity and increased number of propagation paths in three spatial dimensions. Sound attenuation is generally higher compared to 2D simulations. This higher attenuation due to the more accurate representation of the spatial distribution of acoustic energy and its interaction with tissue will lead to a higher temperature increase in the 3D models. This higher temperature increase can be observed by comparing our simulation results with Montienthong & Rattanadecho's results [3]. Although the acoustic frequency is considered the same as Montienthong & Rattanadecho's study and the focal length is approximately the same, higher temperatures at the focal point are achieved in this study in comparison to Montienthong & Rattanadecho simulations. As can be seen in Figs 10 and 11, the temperature rises to 79–80˚C in the current simulation but as shown in Montienthong & Rattanadecho's research (Figs 8 and 9 in their paper) it only rises to 71–72˚C. Moreover, in 3D simulations, the real complex geometry of the breast model with all its details can be implemented. These asymmetric details can significantly impact both acoustic wave propagation and heat transfer phenomena, leading to more accurate results than previously studied 2D axisymmetric models.

In the end, we explained here how the present numerical work will be helpful in addressing current clinical translational research studies. Our 3D FEM-based numerical procedure improves HIFU treatment planning by accurately simulating acoustic fields, thermal distribution, and necrotic tissue formation in realistic breast phantoms. This helps clinicians tailor HIFU parameters to each patient's anatomy, resulting in more effective and personalized treatment plans [64]. This paper emphasizes the importance of customizing HIFU settings based on the patient's anatomy and tumor characteristics. Factors like tumor location, tissue properties, and barriers such as fat or bone can affect treatment effectiveness, challenges that our simulation platform can address. Using patient-specific breast models from T1-weighted MRIs customizes HIFU therapy by allowing clinicians to optimize ultrasound wave parameters. This approach addresses patient anatomy and tumor variability, improving therapeutic outcomes while minimizing damage to healthy tissues [12]. The numerical platform simulates thermal effects and tissue necrosis, accurately predicting temperature and damage. This ensures safer and more effective HIFU treatments by mitigating overheating risks and collateral damage [64]. Validation using previously published experimental data and literature confirms the reliability and accuracy of our model, boosting clinical confidence in 3D simulations for HIFU treatment planning and outcome prediction, and encouraging the adoption of HIFU in clinical practice [65]. Our comprehensive 3D FEM-based simulation approach sets a foundation for future innovations in HIFU therapy. The ability to simulate complex interactions within realistic anatomical structures opens avenues for developing advanced HIFU techniques and integrating them with other therapeutic modalities [23]. This platform can also be used for preclinical testing of new HIFU devices and protocols, fostering continuous improvement in the field. Wu et al. conducted a randomized clinical trial on the use of HIFU for treating localized breast cancer, demonstrating its effectiveness in inducing tumor necrosis with minimal side effects [66]. Similarly, Peek et al. investigated the use of HIFU for treating fibroadenomata, showing significant reductions in lump size and pain, positioning it as a promising non-invasive alternative to surgery [67]. Our numerical platform for HIFU simulations can enhance these clinical findings by optimizing treatment parameters such as energy delivery and targeting accuracy, facilitating the development of efficient, patient-specific treatment strategies while addressing current clinical limitations.

## 5 Limitations and future perspectives

In this section, we discuss some of the limitations of our study and outline potential areas for improvement in future research. These considerations highlight aspects that could be refined to enhance the accuracy and applicability of our findings, offering insights for further investigations.

It is worth noting that the heterogeneity of blood perfusion in the breast plays a critical role in determining the temperature distribution in different parts of the tissue and also in evaluating the subsequent degree of tissue injury. In our simulation platform, the traditional Pennes Bioheat Equation, which is based on the assumption of homogeneous blood perfusion, is considered. As you can see in Eq 3, a constant value of blood perfusion rate ($\omega_b$) is assumed for each tissue type. However, a more realistic assumption would be to consider a spatial distribution of the blood perfusion rate $\omega_b(x,y,z)$, which takes into account the heterogeneity and anisotropy of blood perfusion. Such a modified form of the Pennes Bioheat Equation has been proposed by Singh [68]. This issue can be mentioned as one of the drawbacks of our simulation platform. However, since determining such a function for the blood perfusion rate is challenging and requires experimental measurements, which are not available for human breast tissue at the moment, we are forced to consider only different constant values of blood perfusion rates corresponding to the relevant tissue type.

Also, presence of large blood vessels can significantly impact heat distribution and the outcome of HIFU therapy due to their ability to act as thermal sinks through convective heat transfer, which is critical to consider for accurate treatment planning [69–72]. Our study used the Pennes bioheat transfer equation, which incorporates a perfusion term accounting for the cooling effect of blood flow. While this model is effective for small vessels and capillaries, it does not explicitly consider the significant thermal sinks created by large blood vessels due to convective heat transfer. To address this, future work could include a detailed vascular model by segmenting large blood vessels from MRI data and integrating their thermal effects into the simulation. This would provide a more accurate representation of thermal fields during HIFU treatment, especially for realistic anatomical structures, but would, of course, incur further computational costs.

One important aspect of bioheat modeling is the consideration of thermal relaxation time ($\tau$) in the Dual Phase Lag (DPL) bioheat model, especially for HIFU therapy of breast tumors where intense heat concentrations are involved. Our study does not explicitly incorporate thermal relaxation effects, and the traditional use of the Arrhenius equation does not account for these effects, which is a limitation. However, we have attempted to address this through the integration of a sophisticated bioheat transfer model and detailed temperature-time profiling. In future studies, incorporating the DPL bioheat model to explicitly consider thermal relaxation time will enhance the accuracy of simulations by accounting for the lag in heat transfer response. This is critical for HIFU therapy of breast tumors, as it better simulates the complex heat dynamics involved. Thermal relaxation time in the DPL model represents the delay in heat conduction, capturing the lag between heat application and temperature change, which is particularly important in scenarios with quick temperature changes, such as HIFU therapy. Studies by Bhowmik et al. and Paul et al. highlight the significance of considering thermal relaxation in bioheat models [73, 74].

In living tissues, heating can lead to both reversible and irreversible damage. Cell regeneration at the tumor-healthy tissue interface, supported by oxygen from arterial blood, is crucial for balancing thermal degradation and limiting further damage. Including this self-regeneration of surrounding normal tissues when modeling thermal damage in tumors could be important. By adding a regeneration term to the Arrhenius kinetics formula (Eq 6), this

regeneration of healthy tissues can be accounted for, yielding more accurate and realistic results [75]. However, for simplicity, our simulation platform did not incorporate this modified version of the Arrhenius formula, which is an area that could be improved.

Furthermore, recent research shows that thermal ablation affects the interstitial space, or porosity, of tissues, altering the heating requirements of tumors. The heat from thermal treatment expands this interstitial space and changes the tumor's structure. This modification in porosity can be added to simulations as well. Integrating these dynamic changes into thermal ablation models can improve the precision of treatment protocols, which, for simplicity, have not been considered in our simulation platform.

It is important to mention, recent findings have shown a temperature-dependent delay in thermal damage at 43°C, which suggests that the required heating duration could be reduced by at least 24% when using a modified dynamic thermal damage model compared to traditional Arrhenius models [76]. Our current simulation platform does not include these adjustments, primarily for simplicity. However, incorporating these changes could significantly enhance the accuracy and efficiency of our thermal ablation protocols, potentially reducing treatment times.

One of the key limitations of our model is that, although it provides valuable insights for treatment planning and guidance in determining the applicator trajectory, there is no direct calculation of the applicator trajectory within the simulation. The trajectory of the applicator plays a critical role in HIFU therapy, as an inaccurate treatment path could leave malignant cells untreated or potentially lead to tumor recurrence. Real-world applications, as highlighted in relevant studies such as [77, 78], emphasize the importance of accurately determining the applicator's path to prevent malignant cell regeneration. Future work should focus on integrating trajectory optimization into the simulation model to ensure more precise treatment outcomes.

Additionally, while our model provides a qualitative analysis of necrotic and undamaged tissue volumes for various tumor types, the uneven shape of certain tumors, such as those with rim enhancement or necrotic and hypoxic regions, poses specific challenges. The model does not fully account for the peripheral regions of the tumor, which may have different properties than the core tumor, requiring additional attention during treatment planning. As a result, clinicians may need to adjust their treatment plans based on the tumor's specific characteristics. This challenge is particularly pronounced in cases with rim enhancement or necrotic and hypoxic tumors, where determining whether to include or exclude peripheral tumor regions becomes more complex.

## 6 Conclusions

In this study, we have introduced a comprehensive finite element method (FEM)-based numerical platform to simulate the high-intensity focused ultrasound (HIFU) ablation process specifically tailored for breast tumors. Utilizing a realistic 3D simulation, our platform accurately models the physical phenomena involved in HIFU, providing detailed insights into the acoustic pressure field, sound intensity distribution, temperature elevation, and the subsequent distribution of necrotic tissue.

Validation of our model was rigorously performed by comparing the simulation results with existing experimental data and previous numerical studies. This comparative analysis demonstrated that our 3D simulations more accurately predict the dynamics of acoustic energy distribution and thermal response within the breast tissue. For example, the temperature elevations and the extent of necrotic tissue predicted by our simulations align closely with those observed in clinical outcomes, confirming the reliability of our model.

Moreover, our simulations have highlighted the limitations of traditional 2D models, which often underestimate the temperature profiles and fail to capture the complex interactions between acoustic waves and anatomical structures. By achieving higher temperature elevations and more effective tissue necrosis, our 3D model proved to be superior, particularly in simulating a bowl-shaped transducer with a 0.1 MPa input pulse at 1.5 MHz frequency, which resulted in significant tumor ablation.

Additionally, our platform has been optimized for computational efficiency, crucial for clinical application. This efficiency is achieved through advanced mesh generation techniques and optimized computation algorithms that reduce processing time without compromising the accuracy of the results. This allows for rapid simulation runs, facilitating real-time adjustments and decision-making in clinical settings.

We also explored a more clinically realistic approach by simulating multiple shorter-duration lesions at various tumor locations. This strategy not only achieved a more uniform temperature distribution across the tumor but also enhanced the effectiveness of the necrosis throughout the entire tumor mass, including its margins. The simulation of multiple focal points reflects current clinical practices aimed at comprehensive tumor ablation, addressing challenges such as heat diffusion uncertainty and the heat sink effect due to perfusion.

Incorporating anatomically realistic breast phantoms derived from T1-weighted MRI scans, our platform closely mimics real clinical scenarios, enhancing the predictive accuracy of HIFU treatments. These detailed simulations support tailored HIFU therapy adjustments for individual cases, optimizing treatment efficacy while minimizing damage to surrounding healthy tissue. On the other hand, by investigating various types and shapes of abnormal tissue, we have further illustrated the capabilities of our simulation platform for patient-specific HIFU ablation simulations.

In conclusion, our FEM-based numerical platform not only provides an accurate and reliable method for simulating HIFU ablation but also stands out as a critical tool for advancing the efficacy and safety of breast cancer treatments. The platform's validation against experimental data, combined with its computational efficiency, makes it a formidable resource in the ongoing refinement of HIFU protocols and personalized cancer treatment strategies.

## Supporting information

**S1 File. Simulation parameters.**
(XLSX)

**S2 File. Raw data of figures.**
(XLSX)

## Author Contributions

**Conceptualization:** Reza Rahpeima.

**Data curation:** Reza Rahpeima, Chao-An Lin.

**Formal analysis:** Reza Rahpeima, Chao-An Lin.

**Investigation:** Reza Rahpeima.

**Methodology:** Reza Rahpeima.

**Project administration:** Chao-An Lin.

**Software:** Reza Rahpeima.

**Supervision:** Chao-An Lin.

**Validation:** Reza Rahpeima.

**Visualization:** Reza Rahpeima.

**Writing – original draft:** Reza Rahpeima.

**Writing – review & editing:** Reza Rahpeima, Chao-An Lin.

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
