## [Decision Letter · Decision Letter 0]

21 Feb 2024

PONE-D-24-02187A Comprehensive Numerical Procedure for High-Intensity Focused Ultrasound Ablation of Breast Tumour on an Anatomically Realistic Breast PhantomPLOS ONE

Dear Dr. Lin,

Thank you for submitting your manuscript to PLOS ONE. After careful consideration, we feel that it does not fully meet PLOS ONE publication criteria as it currently stands. Therefore, we invite you to submit a revised version of the manuscript that addresses the points raised during the review process.

We look forward to receiving your revised manuscript.

Kind regards,

Arka Bhowmik, Ph.D.

Academic Editor

PLOS ONE

Journal Requirements:

3. We note that your Data Availability Statement is currently as follows: "All relevant data are within the manuscript and its Supporting Information files"

**Additional Editor Comments:**

Your manuscript has been carefully reviewed by the reviewers and received mixed comments. Please address these mandatory queries (appended here) line-by-line in your Response Letter document with specific details about any changes that were made in your revised manuscript, or the reasons why the suggested changes have not been made.

Reviewers' comments:

Reviewer's Responses to Questions

**Comments to the Author**

1. Is the manuscript technically sound, and do the data support the conclusions?

Reviewer #1: Partly

Reviewer #2: Partly

2. Has the statistical analysis been performed appropriately and rigorously? 

Reviewer #1: No

Reviewer #2: No

3. Have the authors made all data underlying the findings in their manuscript fully available?

Reviewer #1: Yes

Reviewer #2: No

4. Is the manuscript presented in an intelligible fashion and written in standard English?

Reviewer #1: No

Reviewer #2: No

5. Review Comments to the Author

Reviewer #1: Peer Review Report

Manuscript ID: PLoS One (PONE-D-24-02187)

Title: “Comprehensive Numerical Procedure for High-Intensity Focused Ultrasound Ablation of Breast Tumour on an Anatomically Realistic Breast Phantom”

The study “Comprehensive Numerical Procedure for High-Intensity Focused Ultrasound Ablation of Breast Tumour on an Anatomically Realistic Breast Phantom” by Lin et al. lies within the journal scope of PLoS One. The study presented three-dimensional finite element method based numerical procedure to solve a Multiphysics problem using high intensity focused ultrasound (HIFU). The study developed a coupling between acoustic field computed from inhomogeneous Helmholtz equation and temperature field from Pennes Bioheat equation. The study uses an Arrhenius kinetic model to calculate the degree of tissue injury. All simulations were performed using COMSOL Multiphysics software. The breast tumor is assumed to be of spherical shape with 10 mm diameter. The study cannot be accepted in present form and therefore, we propose major revision while giving authors an opportunity to further improve the work while discussing their results. The authors must thrust on logical and quantitative discussion with qualitative inferences, while also considering comparing and contrast with literature. The global perspective was missing, and all dots of the work must connect to conclude meaningful inferences. The authors must state that if they have used any artificial intelligence assisted tools to write, extract or infer any particular section of this work.

1. Blood perfusion heterogeneity of breast have a critical role in ablation treatment effectiveness. The authors used the traditional Pennes Bioheat Equation based on homogeneous or isotropic blood perfusion assumption. However, more recently a modified form of Pennes Bioheat Equation is proposed taking into account heterogeneous or anisotropic blood perfusion [Modified Pennes bioheat equation with heterogeneous blood perfusion: A newer perspective]. This is a drawback of current work since the thermal energy deposition is affected by perfusion heterogeneity in tumors. There are regions of high blood perfusion, moderate perfusion, low perfusion and zero perfusion or necrotic tumor zones. Discuss under discussion or limitations section. This perspective needs to be clearly explained and its relation to tumor hypoxia.

2. Cellular and biological tissue heating may result in reversible (or repairable injury) and irreversible (or lethal) thermal cell-death in living biological tissues. Continuous regeneration of living human tissues due to the continuous supply of oxygen through arterial blood must be taken into account to counter balance the thermal degradation at quasi-static thermal conditions. There is regeneration of healthy cells at the interface of tumor-healthy tissue. At the interface, the regeneration of healthy cells triggers an immune response of biological tissue towards continued heating to suppress, prevent and restrict further accumulation of thermal damage within damage bounds of Ω≤1. While modelling the kinetics of thermal damage of tumour, one must include and should not ignore the partial self-regeneration of connecting normal human tissues at the tumour periphery due to continuous matching of oxygen demands in the healthy tissue by the arterial blood. [Incorporating vascular-stasis based blood perfusion to evaluate the thermal signatures of cell-death using modified Arrhenius equation with regeneration of living tissues during nanoparticle-assisted thermal therapy].

3. Recent studies suggests change in interstitial space (porosity) of tissue during thermal ablation [https://doi.org/10.1016/j.icheatmasstransfer.2021.105393]. Interstitial space of tissue is changed during heating and it also modifies the heating requirements of tumor based on tumor vasculature. Explain and discuss in discussion section?

4. Hyperthermia treatment planning as explained by Soni et al. group involves patient specific model extraction from medical images (MRI/CT), details about size and depth of tumor, type of breast (extremely dense, heterogeneously dense, scattered fibro-glandular and predominantly fatty), estimation of power distribution (specific absorption rate), fringe heating around tumor i.e. sparing healthy tissues around the tumor borders [https://doi.org/10.1016/j.cmpb.2020.105781]. Consider including a figure (pictorial representation) or a text in start of Section-2 in reference to breast cancer study on ablation margins. It will help in determining the role of optimization algorithms in preserving the healthy tissues or minimizing healthy tissues.

5. The study quantified the formation of thermal damage using Arrhenius thermal damage model. One recent study by Zhu and co-workers suggested that there is a temperature dependent time delay at 43°C (Line 201) [Heating Protocol Design Affected by Nanoparticle Redistribution and Thermal Damage Model in Magnetic Nanoparticle Hyperthermia for Cancer Treatment] etc. How would your designed heating protocol would differ for such change of thermal damage metrics? Discuss this issue in the discussion section as the required heating for that published work is atleast 24% shorter than that using the traditional Arrhenius integral?

6. In equation 12, the units of temperature are missing if constant value needed to be reported.

7. One property shouldn’t translate to another or shouldn’t be directly taken from previous published literature such as the coefficient for the heat convection of the breast skin. Justify how did the value of 5 W/m2K is correct for current simulation.

8. The authors should have separate Results and a separate Discussion section.

9. Improve the resolution of figures.

General Comments:

1. Lines 35-36: Extensive editing help is needed for making meaningful inferences.

Among the most prevalent forms of cancer, today is breast cancer, a disease that poses a significant threat to the lives of individuals of both sexes annually.

2. Lines 45-47: Add References to the context.

Although there are alternative thermal ablation techniques such as laser [….], microwave […..], and magnetic fluid hyperthermia (MFH) [Quantitative evaluation of effects of coupled temperature elevation, thermal damage, and enlarged porosity on nanoparticle migration in tumors during magnetic nanoparticle hyperthermia], HIFU is gaining popularity due to several factors [Report Factors here].

3. Lines 203-207 (Language correction needed)

For each physics, it is mentioned what data is required, what the material properties are, and what the initial and boundary conditions are. In the first step, we show the simulation of the ultrasonic wave generation inside the domain. In the second step, simulations of the heat generated and heat transfer inside the breast phantom are carried out.

Lines 386-387 (Language correction needed)

As mentioned before, by using a bowl-shaped acoustic transducer, it is tried to generate a focused ultrasonic wave with a 1.5 MHz frequency at the tumour area.

4. The authors didn’t pose a clear Research Question? What is already known? What is unknown? Why the study is important? How existing approaches have limitations that can potentially be addressed with current study? Was this approach previously tested? If yes, where? If no, why not?

5. Rewrite Conclusion section with meaningful inferences.

6. Discussion should be elaborated by drawing meaningful inferences while comparing and contrasting the true efficacy of the model.

7. One of the primary concerns with this paper is the lack of clarity in defining the scope and objectives of the study. The authors have not clearly outlined the specific research questions or goals they aimed to address. As a result, the paper appears to be a collection of loosely connected ideas rather than a cohesive discussion.

8. Another major issue is the lack of rigorous methodology. The paper does not provide any information regarding the systematic studies used to perform this computation.

9. The paper also suffers from a lack of critical analysis and evaluation of the reviewed technologies. Merely describing and summarizing the existing technologies without offering any meaningful insights or comparisons diminishes the value of the work. The paper should have critically examined the strengths, limitations, and potential areas of improvement for each approach.

10. The overall organization and structure of the paper are inadequate. The flow of ideas is unclear, and there is a lack of coherence between sections. The paper should have presented a clear introduction, outlined the main themes or categories of technologies, and provided a concise summary or conclusion to tie the information together.

11. Spacing issues. Correct such issues throughout the manuscript.

12. Abbreviations must be explained next to the first mention or wherever appropriate to avoid reader struggle understanding the terminology. We have identified technical jargons which must be avoided by the authors.

13. Unrequired abbreviations otherwise should be discarded.

We are looking forward to reviewing your next manuscript with addition of all necessary details recommended by the reviewer.

Reviewer #2: The authors have presented an appreciating simulation work of HIFU ablation of breast tumor models based on realistic anatomical structures. Unfortunately, the novelty of the work in present format is not suitable for publication. I can accept the work for publication if the authors can address the following major comments satisfactorily.

1. Please justify the novelty of the work in clinical aspect. How the proposed HIFU based therapeutic approach is beneficial over the existing non-invasive therapeutic techniques of breast cancer.

2. How the present numerical work will be helpful in solving any current clinically translational research study.

3. Is the HIFU transducer parameters and other operating parameters in present problem has any clinical relevance especially the ablation period.

4. Did the authors considered the effect of large blood vessels and thermal relaxation effect of bio-tissues in present bio-heat model.

5. Realistic breast structures also includes blood vessels, and fluid glands, how this has been addressed in present model.

6. How the proposed numerical model is novel apart from the computational domain.

7. The quantity of work is not suitable for publication.

Minor comments:

1. The total number of figures and tables can be reduced, Figure 1-6, 8 defines same thing i.e. the computational domain; figure 9, 10, 11 are same validation study results. Table 3 and figure 9 conveys similar information.

2. The axis label can be improved especially figure 7, 9, 15.

3. The figure formatting does not look professional, and the resolution is poor, can be improved. Figure 21 seems like a screenshot.

6. PLOS authors have the option to publish the peer review history of their article (what does this mean?). If published, this will include your full peer review and any attached files.

Reviewer #1: **Yes: **Manpreet Singh

Reviewer #2: No

---

## [Author Response · Author response to Decision Letter 0]

16 May 2024

I am writing to submit our revised manuscript entitled, "A Comprehensive Numerical Procedure for High-Intensity Focused Ultrasound Ablation of Breast Tumor on an Anatomically Realistic Breast Phantom," for consideration as a research article in the "PLOS ONE" journal.

We have carefully revised the manuscript in response to the comments provided by the reviewers, which we greatly appreciate. A detailed point-by-point response to the comments is also attached for your reference.

Please let us know if you need any further information.

---

## [Decision Letter · Decision Letter 1]

2 Jun 2024

PONE-D-24-02187R1A Comprehensive Numerical Procedure for High-Intensity Focused Ultrasound Ablation of Breast Tumour on an Anatomically Realistic Breast PhantomPLOS ONE

Dear Dr. Lin,

Thank you for submitting your manuscript to PLOS ONE. After careful consideration, we feel that it has merit but does not fully meet PLOS ONE’s publication criteria as it currently stands. Therefore, we invite you to submit a revised version of the manuscript that addresses the points raised during the review process. Your manuscript has been carefully reviewed by the reviewer. Please address these mandatory queries (appended below) line-by-line in your 'Response Letter' document with specific details about any changes that were made in your revised manuscript, or the reasons why the suggested changes have not been made.

We look forward to receiving your revised manuscript.

Kind regards,

Arka Bhowmik, Ph.D.

Academic Editor

PLOS ONE

Additional Editor Comments:

This paper presents HIFU simulation study on breast 3D model with spherical lesion and fibroglandular tissue. It was emphasized that the critical need is to develop a 3D realistic breast phantom model for HIFU study. I appreciate the volume of work and analysis in this paper. However, my literature search led to another study (10.1371/journal.pone.0274801) using similar 3D model of breast by same group. In this paper, only extension is HIFU simulation of a published model, which is a trivial step for someone working in this field. I believe modifying the applicator, repeating the same analysis as in previous numerical study is not an admirable scientific approach. The paper further failed to receive favorable comments from one of the reviewers. Therefore, I propose following suggestions to improve the scientific contribution of this paper:

1. Introduction:

Although some points were discussed in current introduction, but it lacks logical reasoning of extending HIFU modeling v/s MFH modeling (10.1371/journal.pone.0274801). Therefore, introduction need to add following points as well (a) what are the challenges/problems or shortcoming of previous MFH breast treatment/model?, (b) why MFH breast treatment/model is not sufficient for treatment planning and clinical workup?, (c) how a HIFU model can play a better role in solving these challenges/problems?, (d) what are the gaps/lacunae in previous implementation of HIFU models including their own model, (e) which of these gaps this study is aiming to address. Using this structure, authors could expand the scientific contribution.

2. Critical improvement or voice:

This problem or study lacks originality and critical voice. Breast lesions are not always solid mass. The real-world lesion could be non-mass enhancement and rim enhancing mass (see below figure). Therefore, a critical improvement or voice would be to compare/treatment error/shortcomings of a simulation study on HIFU patient-specific modeling of non-mass enhancement, rim enhancing mass and solid mass. The study can use 3d slicer to construct realistic models of breast with (a) solid mass, (b) non-mass enhancement, and (c) rim enhancing mass, which can be shared as an ‘stl’ file for readers. Authors can use open-source MRI dataset to obtain sample cases of non-mass enhancement, rim enhancing mass and solid mass.

Figure (attached PDF): Sample images of (a) solid mass, (b) non-mass enhancement, and (c) rim enhancing mass.

Open-source database:

https://www.cancerimagingarchive.net/collection/rider-breast-mri/

https://www.cancerimagingarchive.net/collection/acrin-contralateral-breast-mr/

https://www.cancerimagingarchive.net/collection/qin-breast-dce-mri/

Reviewers' comments:

Reviewer's Responses to Questions

**Comments to the Author**

1. If the authors have adequately addressed your comments raised in a previous round of review and you feel that this manuscript is now acceptable for publication, you may indicate that here to bypass the “Comments to the Author” section, enter your conflict of interest statement in the “Confidential to Editor” section, and submit your "Accept" recommendation.

Reviewer #1: All comments have been addressed

Reviewer #2: (No Response)

2. Is the manuscript technically sound, and do the data support the conclusions?

Reviewer #1: Yes

Reviewer #2: Partly

3. Has the statistical analysis been performed appropriately and rigorously? 

Reviewer #1: N/A

Reviewer #2: N/A

4. Have the authors made all data underlying the findings in their manuscript fully available?

Reviewer #1: Yes

Reviewer #2: Yes

5. Is the manuscript presented in an intelligible fashion and written in standard English?

Reviewer #1: No

Reviewer #2: Yes

6. Review Comments to the Author

Reviewer #1: The manuscript is substantially improved following the peer-review comments. There are some minor typographical errors which authors can correct at proof-read stage. For example: the usage of "physic"...Please correct to "physics". I have seen in figure 3 caption, figure 4 caption, Table 2 caption and multiple times in text as noted in line 335. In figure 5, please correct the units W/cm^2. Also, follow consistency as to represent 10^3 as you use in line 403 but refrain to use in other places such as equation 9 or Table 3 or line 537. Please be consistent. Otherwise, the manuscript discussed all needed information.

Reviewer #2: The authors provided a detailed explanation regarding novelty of their work and also revised the manuscript accordingly. It would be satisfactory if the authors can add standard references in support of their explanations provided to justify the novelty of their work. Standard references discussing about limitations of HIFU therapy of breast tumor and clinical challenges which are addressed by present simulation study is missing in the revised manuscript. I would be happy with justification for the novelty of the work if they could provide standard references for every justification. Also simulation studies on HIFU therapy considering 3D realistic anatomical breast phantom are there in literature, below link is an example:

https://jtultrasound.biomedcentral.com/articles/10.1186/s40349-018-0111-9

Carefully site all possible references in support of explanation on how present numerical work will be helpful in solving any current clinically translational research study especially for the points: Improved Treatment Planning, Customization of Therapy, Safety and Efficacy Evaluation, Accelerating Clinical Adoption, Foundation for Future Innovations.

Carefully site all possible references in support of the explanation of clinical relevance of present HIFU transducer and other operating parameters such as ablation period, and ablation parameters.

Accordingly include all standard references in support of your study gap in the revised manuscript, so that the research question addressed in present work and its possible impact to the community is clearly visible.

The drawbacks of present model are more clearly explained in the revised manuscript. However more revision in respective concern is needed especially with the fact that consideration of inhomogeneous blood perfusion rate in bioheat model cannot ignore the effect of large blood vessels.. It is crucial to consider the effect of large blood vessels in bioheat model especially while dealing with realistic anatomical structures.

I am sorry to disagree with the explanation given in the rebuttal regarding consideration of thermal relaxation time in present bioheat model as blood perfusion and thermal conductivity. Literature shows much effort explaining the significance of considering the thermal relaxation time (τ) in Dual phase lag bioheat model especially for the case of HIFU therapy of breast tumor where intense heat concentrations are involved. Authors should consider all possible literature while explaining the limitation of their work.

I also disagree with the explanation on marginal improvement in bioheat model accuracy considering large blood vessels. Literature shows much effort to explain the significance of considering large blood vessels in bioheat model. Blood perfusion rate alone is not sufficient to consider the heat sink effect in bioheat model as in Pennes equation. Authors should clearly justify these facts in their study with appropriate references.

It is not well satisfactory that the novelty of this work extends beyond the incorporation of an anatomically realistic breast phantom in the form of a 3D simulation of HIFU and wave propagation. As there are many literature where HIFU therapy is considered on 3D model and also with realistic breast anatomy. Also proper explanation is needed on how the authors addressed the need of highly dense mesh for a 3D acoustic wave simulation in their FEM model using COMSOL with limited computational resources. Detailed explanation is needed on mesh optimization step, what size of 3D domain is considered where λ/8 mesh size is maintained, justify if the respective domain size is sufficient for a realistic anatomical breast model.

Authors can put more effort in reducing number of figure numbers for a standard publication, Acoustic pressure field and intensity defines similar thing. If the authors want to keep both acoustic intensity and pressure results in the figure then give proper justification. Also what is the need of considering the line graph and 2D map in the figures for acoustic intensity and pressure results justify.

Incorporate line numbers of the revised manuscript to address all above comments within the rebuttal.

7. PLOS authors have the option to publish the peer review history of their article (what does this mean?). If published, this will include your full peer review and any attached files.

Reviewer #1: **Yes: **Manpreet Singh

Reviewer #2: **Yes: **Dr. Abhijit Paul

---

## [Author Response · Author response to Decision Letter 1]

30 Jul 2024

We have carefully revised the manuscript in response to the comments provided by the reviewers, which we greatly appreciate. A detailed point-by-point response to the comments is also attached with this submission.

---

## [Decision Letter · Decision Letter 2]

12 Aug 2024

PONE-D-24-02187R2A Comprehensive Numerical Procedure for High-Intensity Focused Ultrasound Ablation of Breast Tumour on an Anatomically Realistic Breast PhantomPLOS ONE

Dear Dr. Lin,

Thank you for submitting your manuscript to PLOS ONE. After careful consideration, we feel that it has merit but does not fully meet PLOS ONE’s publication criteria as it currently stands. Therefore, we invite you to submit a revised version of the manuscript that addresses the points raised during the review process.

Some minor recommendations are made by the reviewer and me. These can be considered before making a final decision.

We look forward to receiving your revised manuscript.

Kind regards,

Arka Bhowmik, Ph.D.

Academic Editor

PLOS ONE

Journal Requirements:

Additional Editor Comments:

The reviewer made a few small recommendations. Before a decision is made, these suggestions might be incorporated into the document.

1. Some of my and the reviewer's critical comments have been addressed by the authors. Nonetheless, I would reiterate that the parametric numerical analysis pertaining to the 3D HIFU breast model does not constitute a novel contribution. A numerical model's expansion from 2D to 3D cannot be regarded as a novel contribution. Thus, the word "novel" ought to be eliminated from the whole document.

2. The limitations section and other treatment planning challenges pertaining to HIFU should be covered by the authors. The reason is that their model serves as a tool for treatment planning. There is not a qualitative analysis available for various tumor types regarding the volume of necrosis and the volume of undamaged tissue. Because the effective ablation trajectory may not always completely encircle the tumor (resulting in an uneven tumor), the treatment plan may need to include or exclude the peripheral tumor region. When there is rim enhancement and a necrotic or hypoxic tumor, this issue becomes more difficult to handle.

3. The inability to calculate the treatment trajectory of applicator is another drawback of these models (see to the real-world applications 10.1245/s10434-011-2011-x and 10.1016/j.apacoust.2021.108367). The applicator trajectory is crucial in HIFU. An inaccurate trajectory of treatment may cause malignant cells to regenerate.

4. When taking into account rim-enhancing tumor, the abstract should include important findings from their model. This is a crucial information that the abstract should contain.

Reviewers' comments:

Reviewer's Responses to Questions

**Comments to the Author**

1. If the authors have adequately addressed your comments raised in a previous round of review and you feel that this manuscript is now acceptable for publication, you may indicate that here to bypass the “Comments to the Author” section, enter your conflict of interest statement in the “Confidential to Editor” section, and submit your "Accept" recommendation.

Reviewer #2: All comments have been addressed

2. Is the manuscript technically sound, and do the data support the conclusions?

Reviewer #2: Partly

3. Has the statistical analysis been performed appropriately and rigorously? 

Reviewer #2: No

4. Have the authors made all data underlying the findings in their manuscript fully available?

Reviewer #2: Yes

5. Is the manuscript presented in an intelligible fashion and written in standard English?

Reviewer #2: Yes

6. Review Comments to the Author

Reviewer #2: I appreciate the author’s effort in defending respective comments and accordingly addressed all comments satisfactorily. Additionally the following points need to clarify and revise for improvement.

1. In response to 1st comment from authors: “Recognizing this limitation, our study introduces a novel all-in-one 3D FEM-based simulation approach using an anatomically realistic breast phantom”. The authors should carefully revise the term ‘novel work’ in the whole manuscript as the literature shows works on MRI based 3D modeling of anatomically realistic breast tissue.

2. I am not fully satisfied with the response to of 2nd comment: ‘carefully site all possible references in support of explanation on how present numerical work will be helpful in solving any current clinically translational research study’,

a. one of the authors response is ‘This helps clinicians tailor HIFU parameters to each patient's anatomy, resulting in more effective and personalized treatment plans [59] Li F, Feng R, Zhang Q, Bai J, Wang Z. Estimation of HIFU induced lesions in vitro: Numerical simulation and experiment. Ultrasonics. 2006;44:e337-e40’, please justify how the paper presents a current clinical challenge, also the referred paper is not a clinical study, it’s an in-silico and in-vitro study, and there is no patient anatomy related information.

b. I appreciate referring clinical studies in response to 2nd comment. In response the authors referred one review article and others are in-silico and in-vitro and ex-vivo studies.

3. Also the authors need to revise this line ‘Validation against experimental data and literature confirms our model's reliability and accuracy’ as it can create confusion to the reader that the ‘experimental work’ is performed in present manuscript.

7. PLOS authors have the option to publish the peer review history of their article (what does this mean?). If published, this will include your full peer review and any attached files.

Reviewer #2: **Yes: **Abhijit Paul

---

## [Author Response · Author response to Decision Letter 2]

7 Sep 2024

We have revised the manuscript according to the comments by the reviewer and editor and the responses are attached with this submission.

---

## [Editor Report · Decision Letter 3]

9 Sep 2024

A Comprehensive Numerical Procedure for High-Intensity Focused Ultrasound Ablation of Breast Tumour on an Anatomically Realistic Breast Phantom

PONE-D-24-02187R3

Dear Dr. Lin,

We’re pleased to inform you that your manuscript has been judged scientifically suitable for publication and will be formally accepted for publication once it meets all outstanding technical requirements.

Kind regards,

Arka Bhowmik, Ph.D.

Academic Editor

PLOS ONE

---

## [Editor Report · Acceptance letter]

19 Sep 2024

PONE-D-24-02187R3 

PLOS ONE

Dear Dr. Lin, 

I'm pleased to inform you that your manuscript has been deemed suitable for publication in PLOS ONE. Congratulations! Your manuscript is now being handed over to our production team.

Kind regards, 

on behalf of

Dr. Arka Bhowmik 

Academic Editor

PLOS ONE